# Fluorinated chlorin chromophores for red-light-driven $CO_2$ reduction

Shuang Yang[1], Huiqing Yuan[1], Kai Guo [1], Zuting Wei[1], Mei Ming[1,2], Jinzhi Yi[1], Long Jiang [1] & Zhiji Han [1] ✉

The utilization of low-energy photons in light-driven reactions is an effective strategy for improving the efficiency of solar energy conversion. In nature, photosynthetic organisms use chlorophylls to harvest the red portion of sunlight, which ultimately drives the reduction of $CO_2$. However, a molecular system that mimics such function is extremely rare in non-noble-metal catalysis. Here we report a series of synthetic fluorinated chlorins as biomimetic chromophores for $CO_2$ reduction, which catalytically produces CO under both 630 nm and 730 nm light irradiation, with turnover numbers of 1790 and 510, respectively. Under appropriate conditions, the system lasts over 240 h and stays active under 1% concentration of $CO_2$. Mechanistic studies reveal that chlorin and chlorinphlorin are two key intermediates in red-light-driven $CO_2$ reduction, while corresponding porphyrin and bacteriochlorin are much less active forms of chromophores.

Light-driven reduction of $CO_2$ ($CO_2$RR) into valuable chemicals has been considered as a promising approach in the direct utilization of solar energy[1–6]. Great progress has been achieved in the study of photochemical $CO_2$RR using the high-energy portion of sunlight[7–17]. However, the development of catalytic systems performing under the irradiation of low-energy light (red and near-infrared) remains a significant challenge. Under AM1.5G, the maximum harvestable photons below 600 nm is ~19%[18]. Thus, there is increasing enthusiasm in seeking photocatalytic systems for the utilization of low-energy photons in the solar spectrum.

Chromophores that have been demonstrated to utilize low-energy photons for catalytic $CO_2$RR are rarely reported in the literature, all of which are based on the most precious metals. A frequent difficulty is that molecules absorbing at long wavelengths often generate insufficient reduction potentials to overcome the large driving forces in $CO_2$RR. In 2013, Ishitani and co-workers reported the first molecular system for red-light-driven $CO_2$RR using Os–Re supramolecular complexes, achieving a turnover number (TON) of 1138 with >620 nm light[19]. Recently, a photochemical system using a heteroleptic Os(II) chromophore and a Ru(II) catalyst was developed for $CO_2$RR to generate HCOOH under irradiation with 725 nm light (TON = 81)[20].

Furthermore, a Zn porphyrin-sensitized Mn(I) system was demonstrated to reduce $CO_2$ under 625 nm light, however, its $TON_{CO}$ was lower than 1[21].

Porphyrin-based compounds have been widely used as blue-light absorbers in artificial photosynthetic systems, either for the reductive-half[22–30] or for the overall reactions[31–33]. However, chlorin (Ch), which is a reduced porphyrin adapted by most of the chlorophylls in plants and cyanobacteria (Fig. 1) for harvesting red-light[34–37], has not been reported in such a context. The Ch can undergo a $2e^-/2H^+$ reduction photochemically to generate a chlorinphlorin (ChPh)[38–43], which exhibits a broad absorption spectrum into the near-infrared region[44–46], as recently determined by Nocera and co-workers[47]. The optical and redox properties of Chs highly suggest that they may serve as active chromophores for red-light-driven $CO_2$RR. Here, we present a series of fluorinated Chs (Fig. 1) as chromophores for red-light-driven $CO_2$RR in precious-metal-free systems. The structure-function study demonstrates that increasing the number of fluorine substituents on the *meso*-phenyl group of Ch significantly enhances the activity of $CO_2$RR. The systems using a per-fluorinated Ch last over 240 h and give high TONs of 1790 (at 630 nm) and 510 (at 730 nm) in the conversion of $CO_2$ to CO.

[1]MOE Key Laboratory of Bioinorganic and Synthetic Chemistry, School of Chemistry, IGCME, Sun Yat-sen University, Guangzhou, China. [2]School of Materials Science and Engineering, Xihua University, Chengdu, China. ✉e-mail: hanzhiji@mail.sysu.edu.cn

**Fig. 1 | Structure diagram.** Structures of chlorophyll *a* in nature, chromophores $F_xTPP$, $F_xCh$ (x = 0, 4, 12, 20), and $F_{20}BC$, catalyst FeTDHPP, and electron donor BIH in the study.

**Table 1 | Photophysical, electrochemical, and photocatalytic $CO_2$ reduction data of $F_xCh$**

| | λ (nm) ε (×$10^4$ $M^{-1}$ $cm^{-1}$) | $E_{red}^a$ (V vs. SCE) | $E_{red}^b$ (V vs. SCE) | $TON^c$ (630 nm) | $TOF^c$ ($h^{-1}$) | TON ($F_xCh$)$^c$ /TON ($F_xTPP$)$^d$ | $TON^e$ (730 nm) |
|---|---|---|---|---|---|---|---|
| $F_0Ch$ | 650 (3.56) | −1.11, −1.58 | −1.05, −1.81 | 210 ± 42 | 8 | 52.5 | 36 |
| $F_4Ch$ | 651 (3.46) | −1.01, −1.52 | −0.98, −1.82 | 558 ± 16 | 112 | 3.7 | 64 |
| $F_{12}Ch$ | 653 (4.64) | −0.85, −1.40 | −0.86, −1.33, −1.78 | 740 ± 78 | 26 ± 2 | 1.4 | 68 |
| $F_{20}Ch$ | 654 (5.13) | −0.73, −1.24 | −0.72, −1.15, −1.61 | 1790 ± 52 | 194 ± 9 | 1.0 | 510 |

Error bars denote standard deviations, based on at least three separated runs. TON and TOF calculated based on [FeTDHPP].
[a]Under $N_2$.
[b]Under $CO_2$ with 1% $H_2O$.
[c]50 μM $F_xCh$, 1.0 μM FeTDHPP, and 50 mM BIH, λ = 630 nm (110 mW/$cm^2$), TON calculated in 51 h, TOF calculated in 15 h for $F_0Ch$, 3 h for $F_4Ch$, 27 h for $F_{12}Ch$, and 6 h for $F_{20}Ch$.
[d]50 μM $F_xTPP$, 1.0 μM FeTDHPP, and 50 mM BIH, λ = 630 nm (110 mW/$cm^2$), TON calculated in 75 h.
[e]50 μM $F_xCh$, 1.0 μM FeTDHPP, and 50 mM BIH, irradiated at λ = 630 nm (110 mW/$cm^2$) for 5 min then λ = 730 nm (80 mW/$cm^2$) for 170 h. Source data are provided as a Source data file.

## Results and discussion

### Synthesis and photophysical properties

A library of fluorinated *meso*-tetraphenyl porphyrins ($F_xTPP$) and chlorins ($F_xCh$) (x = 0, 4, 12, 20; Fig. 1) were prepared according to procedures described in the "Methods" section. A per-fluorinated bacteriochlorophyll ($F_{20}BC$, Fig. 1) was synthesized by further reduction of $F_{20}Ch$ and crystallographically determined (Supplementary Information). In contrast to $F_xTPP$, all $F_xCh$ display strong absorption profiles for red light in N,N-dimethylformamide (DMF) (Supplementary Figs. 2–6). Increasing the number of the fluorine substituents on $F_xCh$ shows a general impact on the absorption band at the red-light region, by slightly red-shifting the maximum absorption from 650 to 654 nm, as well as increasing the extinction coefficient (ε) from 3.458 × $10^4$ to 5.125 × $10^4$ $M^{-1}$ $cm^{-1}$ (Table 1). All chlorins exhibit an intense Soret (B) bond (from 405 to 420 nm) and 4 to 5 Q bands (from 504 to 654 nm). As would normally be anticipated for chlorins[48–51], red-shift

and the higher extinction coefficient for the Q band between 650 and 654 nm than those observed for corresponding $F_xTPP$.

### Light-driven $CO_2$RR

The light-driven properties of these macrocyclic chromophores were evaluated in a well-studied $CO_2$RR system in our laboratory, by using an iron (III) tetrakis (2′,6′-dihydroxyphenyl)-porphyrin (FeTDHPP) as the catalyst and 1,3-dimethyl-2-phenyl benzimidazoline (BIH) as the electron donor[52,53]. The system containing these three components in a $CO_2$-saturated DMF solution was irradiated using a red light-emitting diode ($λ_{max}$ = 630 or 730 nm, Supplementary Fig. 8), and the gaseous products generated were measured in real-time by gas chromatography (GC).

We first examined the activity of a simple *meso*-tetraphenyl porphyrin $F_0TPP$ for red-light-driven $CO_2$RR. In a system (50 μM $F_0TPP$, 1.0 μM FeTDHPP, 50 mM BIH, 630 nm light), only a small amount

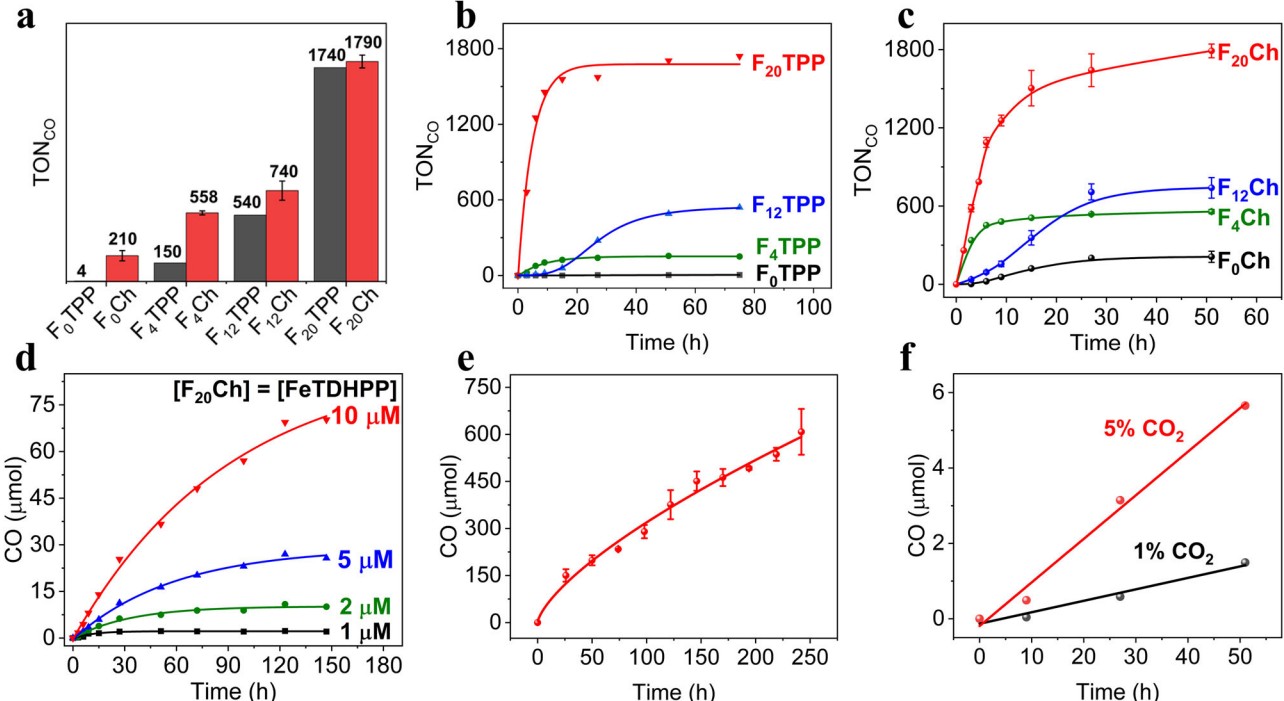

**Fig. 2 | Photochemical data of CO$_2$RR. a** Comparing the overall TON$_{CO}$ of systems with different F$_x$TPP and F$_x$Ch. **b** TON$_{CO}$ for systems with different F$_x$TPP. **c** TON$_{CO}$ for systems with different F$_x$Ch. **d** CO generation for systems with different initial [F$_{20}$Ch] and [FeTDHPP]. **e** stability of a system with F$_{20}$Ch, **f** CO$_2$RR under 1% or 5% concentrations of CO$_2$ with F$_{20}$Ch. Error bars denote standard deviations. TON calculated based on [FeTDHPP]. Catalytic conditions: **a–c** used 50 μM F$_x$TPP or F$_x$Ch, 1.0 μM FeTDHPP, and 50 mM BIH; **d** used the same concentrations (1, 2, 5,

10 μM) of F$_{20}$Ch and FeTDHPP, 50 mM BIH; **e** used 100 μM F$_{20}$Ch, 100 μM FeTDHPP, and 200 mM BIH; **f** used 100 μM F$_{20}$Ch, 100 μM FeTDHPP, and 50 mM BIH; Experiments were in CO$_2$-saturated DMF (5.0 mL) at 20 °C using a light-emitting diode (LED) source (λ = 630 nm, 110 mW/cm$^2$). The profiles and data of TOF for (**b–e**) were shown in Supplementary Fig. 9, Table 1, and Supplementary Tables 4 and 5. Source data are provided as a Source data file.

(0.02 μmol) of CO corresponding to a TON of 4.0 was detected. We found that replacing F$_0$TPP with F$_0$Ch in the system resulted in a >52-fold increase of TON at 630 nm (Fig. 2a). A remarkable observation for both F$_x$TPP and F$_x$Ch is the increases in CO$_2$RR activity with more fluorine substituents on the chromophore (Fig. 2a–c, and Table 1). With the per-fluorinated chlorin F$_{20}$Ch, the system achieves a high TON of 1790 ± 52 after 51 h of irradiation, an initial turnover frequency (TOF) up to 194 ± 9 h$^{-1}$, and a quantum yield of 0.88 ± 0.03% at 630 nm (Supplementary Table 3). The TON described here is significantly higher than those reported for red-light-driven CO$_2$RR (homogeneous and heterogeneous) systems including the ones using noble metals (Supplementary Tables 1 and 2).

We further found that the initial rates of CO production followed a first-order dependence on both concentrations of chromophore and catalyst (Supplementary Figs. 11 and 12). Based on the observation, we hypothesized that high TONs for both chromophore and catalyst could be realized in one system, which would be beneficial for the development of versatile light-driven and light-electricity-driven systems. Indeed, when the CO$_2$RR experiments were conducted under the same concentration of F$_{20}$Ch and FeTDHPP, a high TON of 1404 was obtained after 147 h (Supplementary Table 4).

In all the red-light-driven experiments performed under an atmosphere of CO$_2$, no other gaseous product was observed by GC. Analysis of the liquid phase by $^1$H NMR showed no detection of formic acid and methanol. To study the selectivity of the system further, we found that the amounts of CO generated were near the theoretical maximum value (Supplementary Fig. 13) of the electron donor BIH (based on two electrons per BIH molecule), which suggests that the selectivity of CO is nearly 100%.

We note that both the stability of the system and the amount of produced CO are significantly improved at higher concentrations of

F$_{20}$Ch and FeTDHPP (Fig. 2d). To study stability and scalability of the system further, we found that the photocatalysis lasted over 240 h and produced over 608 μmol CO when the experiments were conducted at 0.1 mM F$_{20}$Ch and FeTDHPP (Fig. 2e). The slight decrease of the initial catalytic rate is presumably due to consumption of CO$_2$ during CO$_2$RR. Indeed, we observed slower rates of CO generation from mixtures at lower concentrations of CO$_2$ (Fig. 2f). However, its ability to function at low CO$_2$ contents (down to 1%) with high selectivity (97.7% under 5% CO$_2$ and 95.6% under 1% CO$_2$) was impressive.

To study the nature of the system, various control experiments were performed. We observed no CO generated from a light-driven system carried out under an atmosphere of N$_2$, which implied that CO was derived from CO$_2$. Isotopic labeling experiments conducted under $^{13}$CO$_2$ produced $^{13}$CO as detected by GC-MS (Supplementary Fig. 14). This result thus further confirms that CO$_2$ is the carbon source in catalysis. In addition, a negligible amount of CO detected in the absence of F$_x$Ch, FeTDHPP, BIH, light, or under Ar (Supplementary Table 6) suggests that all components are essential for the light-driven CO$_2$RR. To rule out potential metal contaminants, inorganic salts (Fe$^{3+}$, Cu$^{2+}$, Ni$^{2+}$, Co$^{3+}$, Ru$^{3+}$, Pd$^{2+}$) with or without TDHPP ligand all produced no or negligible amount of CO as compared with the experiment using FeTDHPP as the catalyst (Supplementary Table 7). Photolysis performed using chemicals that passed the elemental analysis or using FeTDHPP synthesized from highly pure FeBr$_2$ all showed identical activity (Supplementary Fig. 15). Furthermore, experiments conducted in the presence of an excess amount of Hg$^0$ showed an identical activity profile (Supplementary Fig. 16), indicating no contamination from amalgam-forming metals. Dynamic light-scattering measurements showed the absence of nanoparticles in the CO$_2$RR systems before and after light irradiation (Supplementary Fig. 17).

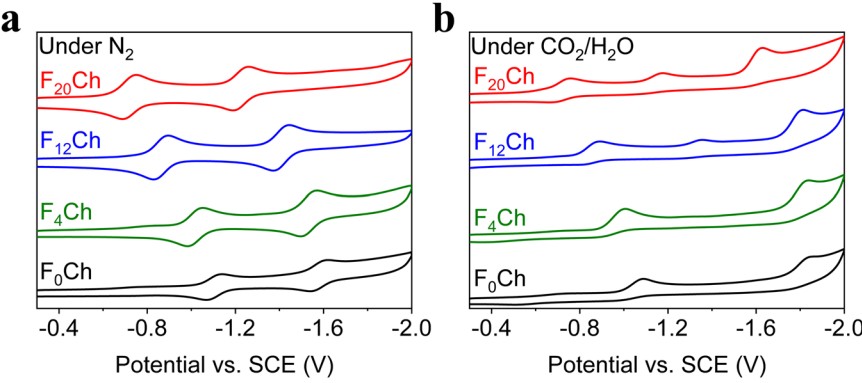

**Fig. 3 | Electrochemical data.** Cyclic voltammograms of 0.25 mM $F_0Ch$, 0.5 mM $F_4Ch$, 0.5 mM $F_{12}Ch$, and 0.5 mM $F_{20}Ch$ in DMF containing 0.1 M TBAPF$_6$ at scan rate 0.1 V/s. **a** Under $N_2$. **b** Under $CO_2$ with 1% $H_2O$. Source data are provided as a Source data file.

## Mechanistic study

To gain mechanistic insight into the red-light-driven system, we next sought to identify the active intermediates in $CO_2RR$. Under Ar or $N_2$, the cyclic voltammogram (CV) of $F_0Ch$ in DMF shows two reversible redox events at −1.11 and −1.58 V vs SCE (Fig. 3a, Supplementary Fig. 18, and Table 1), corresponding to the generation of one and two electrons reduced Chs ($F_0Ch^-$ and $F_0Ch^{2-}$)[39]. For the fluorinated Chs, these two reduction potentials shift towards more positive (by over 300 mV for $F_{20}Ch$) due to the electron-withdrawing effects of the fluorine substituents (Fig. 3a, Supplementary Fig. 19, and Table 1). Similar CV spectra were obtained in experiments conducted in the presence of 1% $H_2O$ (Supplementary Fig. 20). Because $CO_2RR$ has to occur at an Fe(0) state of FeTDHPP at −1.55 V vs SCE[54–56], a more reducing form of chromophore other than $F_xCh^-$ and $F_xCh^{2-}$ must be involved in the photochemical scheme. Indeed, the CV, obtained under an atmosphere of $CO_2$ and in the presence of $H_2O$ as the proton source, revealed the appearance of a new reduction wave at a potential more negative than −1.6 V (Fig. 3b, Supplementary Fig. 21, and Table 1). A similar observation was obtained when using either trifluoro ethanol or acetic acid as the proton donor (Supplementary Figs. 22–26). In these experiments, the increase of the first reduction wave and the decrease of the second reduction wave suggests the generation of $F_xChPh$ (Fig. 4) through two subsequent proton-coupled electron transfer[47]. Therefore, the wave at <−1.6 V is ascribed to further reduction of $F_xChPh$ to $F_xChPh^-$ (Fig. 4), which is thermodynamically favorable in reducing FeTDHPP to the required Fe(0) intermediate for $CO_2RR$.

We gained further evidence of the $F_xChPh$ species using ultraviolet-visible (UV−vis) spectroscopy. By photolyzing a solution containing $F_0Ch$ and BIH for a few minutes, we observe a broad absorption peak (from ~450 nm to 850 nm) that corresponds to a 2e$^-$/2H$^+$ photoproduct $F_0ChPh$ (Supplementary Fig. 27)[39,46,47]. Consistent with this result, similar broad spectra are observed in the reduction of other $F_xCh$ (Supplementary Figs. 28–30). In addition, most of the $F_0Ch$ can be recovered in a reverse 2e$^-$/2H$^+$ oxidative process by exposing the photolyzed solution to the air (Supplementary Fig. 27c).

We next examined the UV−vis spectra of the catalytic solutions (containing $F_0Ch$ or $F_{20}Ch$, FeTDHPP, and BIH) during photolysis (Supplementary Figs. 31 and 32). The broad spectra corresponding to $F_xChPh$ were again quickly generated within minutes. Continued irradiation resulted in a slow decrease of $F_xChPh$ during $CO_2RR$, suggesting a further reduction of $F_xChPh$ to $F_xChPh^-$. By keeping the photolyzed $F_{20}Ch$-containing system in the dark, we observed recovery of $F_{20}ChPh$ and generation of $F_{20}Ch$ and $F_{20}BC$ in the solution (Supplementary Fig. 33), as well as an additional 0.66 ± 0.03 equiv of CO in the headspace (Supplementary Table 8). These findings thus suggest that both photochemical conversion of $F_xChPh$ to $F_xChPh^-$ (through reductive quenching) and electron transfer from $F_xChPh^-$ to FeTDHPP (to generate the Fe(0) intermediate) are the two key steps in $CO_2RR$ (Fig. 4).

We performed additional experiments to study whether $F_xChPh^-$ is also involved in red-light-harvesting in $CO_2RR$. Because it exhibits absorption up to ~600 nm (Supplementary Figs. 31 and 32), the corresponding photochemical pathway in $CO_2RR$ should be terminated by using a light source with longer wavelengths. We found that all four $F_xCh$ were active in producing CO over 170 h under irradiation with 730 nm LEDs (Table 1 and Supplementary Fig. 34). In the series, the system with $F_{20}Ch$ gave the highest $TON_{CO}$ of 510. This suggests that $F_xChPh$ instead of $F_xChPh^-$ is responsible for absorbing red light in $CO_2RR$.

To evaluate the stability of $F_xCh$ during $CO_2RR$, we quenched the photolysis by treatment of the catalytic solution first using a Co(III) dimethylglyoximate complex and then exposure to the air. For the $F_0Ch$-containing system, UV−vis spectra revealed a 71% decrease of $F_0Ch$ and a significant increase of the $F_0BC$ after 75 h irradiation (Supplementary Fig. 35). In comparison, 51% of $F_{20}Ch$ was recovered and a relatively small amount of $F_{20}BC$ was observed (Supplementary Fig. 36). These results imply that cessation of $CO_2RR$ under these conditions may be due to complete decomposition of FeTDHPP. In fact, at a higher [FeTDHPP], the lifetime of the system is significantly prolonged (Fig. 2d, e), as described above. Furthermore, addition of FeTDHPP to a photolysis system at 51 h completely restored the activity (Supplementary Fig. 37), which confirms that deactivation of FeTDHPP is a limiting factor in the lifetime of the system.

To understand the very different activity between $F_xTPP$ and $F_xCh$, we examined the photochemical steps of $F_xTPP$ in $CO_2RR$. For the least active chromophore $F_0TPP$, no conversion to $F_0Ch$ was observed during photolysis (Supplementary Fig. 38). In its quenched reaction mixture, we found that most of the $F_0TPP$ was recovered after 4 h irradiation (Supplementary Fig. 39). However, for other fluorinated TPP, $F_xCh$ intermediates can be observed unambiguously (Supplementary Figs. 40–42). Furthermore, a significant conversion of $F_{20}TPP$ to $F_{20}Ch$ was observed (Supplementary Fig. 43), which might explain the much higher activity observed with $F_{20}TPP$ in the series. This evidence suggests that the fluorine substituents on TPP facilitate isomerization from phlorin (a 2e$^-$/2H$^+$ reduced porphyrin, defined as $F_xPh$) to Ch (Fig. 4), and such transformation is an essential pathway when using $F_xTPP$ as the chromophore for red-light-driven $CO_2RR$.

Because $F_xBC$ (presumably isomerized from $F_xChPh$) is present in the quenched photolysis mixtures, we study its impact on $CO_2RR$ with an independently synthesized $F_{20}BC$. The crystal structure of $F_{20}BC$ shows two characteristic C−C single bonds in the pyrrole ring, which

**Fig. 4 | Proposed mechanism for the red-light-driven CO₂RR.**

exhibit similar distances compared with reported Ch and bacterio-chlorins (BC) compounds (Supplementary Table 10)[57,58]. Under the same conditions, the system with $F_{20}BC$ exhibits a much lower initial TOF (60 h⁻¹) than that with $F_{20}Ch$ (Supplementary Fig. 44). UV–vis study shows no detection of $F_{20}Ch$ in both the photocatalytic and the quenched solutions (Supplementary Figs. 45 and 46). Hence, the irreversible isomerization from $F_xChPh$ to $F_xBC$ (Fig. 4) during photolysis might also lead to a decrease of CO₂RR activity when using $F_xCh$ as the chromophores.

Previous studies showed that BIH functioned as a 2e⁻/1H⁺ donor[59,60]. To generate the BI-radical (which donates the second e⁻), deprotonation of the BIH-radical cation by a base such as triethylamine (TEA) was found to be necessary in acetonitrile (ACN) (Supplementary Table 11). However, there are several photocatalytic studies reported in DMF without TEA or additional bases when using BIH as the electron donor[53,61,62]. To investigate this, we performed CV studies for BIH in DMF and ACN (Supplementary Figs. 47–48). In contrast to the voltammograms in ACN, the CV in DMF showed appearance of a reduction wave at ~ −1.6 V vs SCE, corresponding to generation of the BI-radical. This result suggests that deprotonation of the BIH-radical cation is more favorable in DMF than in ACN. However, addition of TEA to the system was found to improve the activity (Supplementary Fig. 49 and Supplementary Table 11), exhibiting a $TON_{CO}$ of 2132 in 27 h and an initial $TOF_{CO}$ of 584 h⁻¹. In the overall reactions, BI⁺ and OH⁻ were produced in generation of the 2e⁻/2H⁺ reduced chromophores and in CO₂ reduction (Eqs. 1 and 2).

$$BIH + CO_2 \rightarrow BI^+ + OH^- + CO \qquad (1)$$

$$BIH + PS + H_2O \rightarrow BI^+ + OH^- + PS - H_2 \qquad (2)$$

The photocatalytic CO₂ reduction mechanism by FeTDHPP has been extensively investigated by Robert and co-workers[63–65]. UV–vis studies showed generation of the corresponding Fe(II) and Fe(I) species at the early stage of photolysis (Supplementary Figs. 50–53). The Fe(I) species was found to decrease during CO production, which indicates a catalytic cycle consistent with previous reports (Fig. 4)[52,64,65]. No electrostatic interaction was found between the chlorin and FeTDHPP by UV–vis studies (Supplementary Fig. 54), which suggests electron transfer from the chromophore to the catalyst follows an outer-sphere mechanism.

Overall, red-light-driven reduction of CO₂ was achieved using a series of synthetic porphyrin-based chromophores in precious-metal-free systems. Conversion of TPP to Ch and ChPh has been identified as an important photochemical pathway in CO₂RR. Fluorination of the light-harvesting macrocycle has been demonstrated to be an effective method both in facilitating such transformation and in promoting the catalytic activity. In light of the high TON, long-term stability, and selectivity of the systems, we anticipate that this study maps a route for the development of efficient CO₂RR systems using low-energy sunlight.

## Methods
### Materials
All solvents and reagents were commercially purchased and used as received without further purification unless otherwise noted. FeTDHPP was prepared following a reported procedure using $FeCl_2 \cdot 4H_2O$ and $FeBr_2$[56], analysis (calcd., found for $C_{44}H_{28}ClFeN_4O_8 \cdot 1.7H_2O \cdot C_6H_{14}$(Hex)): C (63.29, 63.39), H (4.82, 5.11), N (5.90, 5.70).

DMF (Energy chemical, 99.8%, distilled, extra dry with molecular sieves, water ≤50 ppm (by K.F.)) was purchased from Anhui Zesheng Technology Co., Ltd (Anuhi, China); Anhydrous $K_2CO_3$ (GREAGENT, ≥99.0%) was purchased from Guangzhou beier biological Technology Co., Ltd (Guangzhou, China); p-Toluenesulfonyl hydrazide (Macklin, 98%) was purchased from Guangzhou Sopo biological Technology Co., Ltd (Guangzhou, China); Dry pyridine (Acseal, 99.5%, with molecular sieves, water ≤50 ppm (by K.F.)) was purchased from Shanghai Jizhi Biochemical Technology Co., Ltd

(Shanghai, China); 2,3-Dichloro-5,6-dicyano-benzoquinone (DDQ, Energy chemical 98%) was purchased from Anhui Zesheng Technology Co., Ltd (Anuhi, China); Benzaldehyde (innochem, 98%); 2-Fluorobenzaldehyde (BIDE, 98%); 2,4,6-Trifluorobenzaldehyde (BIDE, 98%); Pentafluorobenzaldehyde (macklin, 98%); 2,6-Dimethoxybenzaldehyde (BIDE, 98%); Boron trifluoride diethyl etherate (Aladdin, $BF_3$: 46.5%); Pyrrole (Energy chemical, 99%); $Co(NO_3)_3 \cdot 6H_2O$ (damas-beta, 99.0%); $Fe(NO_3)_3 \cdot 9H_2O$ (Guangzhou, ≥98.5%); $Cu(NO_3)_2 \cdot 9H_2O$ (Kermel, 99.0–102.0%); $Ni(NO_3)_2 \cdot 6H_2O$ (Xiya, 99%); $RuCl_3 \cdot xH_2O$ (Aladdin, 35.0–42.0% Ru basis); $FeCl_2 \cdot 4H_2O$ (Guangzhou, 99.5–101.0%); $FeBr_2$ (Rhawn, 99.995% metals basis); $Pd(OAc)_2$ (Eybridge, 98%); Dichloromethane (WOHUA-CHEMICAL, 99.5%); Petroleum ether (PE, WOHUA-CHEMICAL, 99.5%); Ethyl acetate (WOHUA-CHEMICAL, 99.5%); n-Hexane (HD, 99.5%).

### Preparation of Co(dmgH)₂PyCl

Co(dmgH)₂PyCl was synthesized following a modified procedure based on previous report[66]. $CoCl_2 \cdot 6H_2O$ (2.15 mmol, 500 mg) was dissolved in 200 mL ethanol and heated to 70 °C, then dimethylglyoxime (4.70 mmol, 551 mg) was added. After 10 min of stirring, pyridine (4.30 mmol, 344 mg) was added drop by drop to the mixture and air was bubbled through the solution for 30 min. The yellow precipitate was collected by filtration and washed with deionized water, ethanol, and diethyl ether, and dried under vacuum. Large yellow block crystals were obtained from acetonitrile by slow evaporation at ambient temperature (75% yield). Co(dmgH)₂PyCl was evidenced by [1]H NMR (Supplementary Fig. 86).

### Preparation of BIH

BIH was prepared based on modified methods in the literature[67]. 2-Phenylbenzimidazole (30.91 mmol, 6.00 g) was dissolved in 30 mL methanol solution containing NaOH (32.00 mmol, 1.28 g), then methyl iodide (112.38 mmol, 7 mL) was added to the above solution and the mixture was heated at 100 °C for 24 h in the dark. After cooling to room temperature, the faint yellow solid ($BIH^+I^-$) was collected by filtration and washed with EtOH/H₂O (5:1, v/v). Then, a solution of $BIH^+I^-$ (8.57 mmol, 3.00 g) in methanol (80 mL) was added slowly with NaBH₄ (89.47 mmol, 3.40 g) under N₂ and the mixture was allowed to react for 3 h. The resulting solution was evaporated to obtain a white solid. The white solid (BIH) was purified by washing with plenty of water. The yield was 90%. BIH was evidenced by [1]H NMR (Supplementary Fig. 84) and elemental analysis (calcd., found for $C_{15}H_{16}N_2$): C (80.32, 80.20), H (7.19, 7.26), N (12.49, 12.42).

### Preparation of F₀TPP

F₀TPP was prepared according to a method in the literature[68]. Pyrrole (0.36 mol, 25 mL) and benzaldehyde (0.38 mol, 40 mL) were dissolved in propionic acid (250 mL) and refluxed for 45 min. When the resulting solution was cooled to ambient temperature, a purple solid was collected by filtration, washed with methanol, and dried under vacuum. The yield was 80%. F₀TPP was evidenced by [1]H NMR (Supplementary Fig. 66) and elemental analysis (calcd., found for $C_{44}H_{30}N_4 \cdot 0.8H_2O$): C (84.00, 84.00), H (5.06, 5.07), N (8.91, 8.81).

### Preparation of F₄TPP

F₄TPP was synthesized based on modified methods in the literature[69]. Pyrrole (0.05 mol, 3.355 g) and 2-fluorobenzaldehyde (0.05 mol, 6.205 g) were added dropwise simultaneously to a boiling propionic acid (200 mL) and the mixture was refluxed for another 30 min. When the resulting solution was cooled to ambient temperature, a purple product (16% yield) was obtained by filtration, and washed with methanol then dried under vacuum. F₄TPP was evidenced by [1]H NMR, [19]F NMR (Supplementary Figs. 67 and 68) and elemental analysis (calcd., found for $C_{44}H_{26}F_4N_4 \cdot 0.3H_2O$): C (76.36, 76.49), H (3.87, 4.18), N (8.10, 8.06).

### Preparation of F₁₂TPP and F₂₀TPP

F₁₂TPP and F₂₀TPP were synthesized according to modified methods from the literature[70]. 2,4,6-Trifluorobenzaldehyde (13.00 mol, 2.080 g) or pentafluorobenzaldehyde (13.00 mol, 2.548 g) was dissolved in 500 mL dichloromethane (DCM), followed by addition of pyrrole (13.00 mmol, 905 μL). After the mixture was stirred and degassed by N₂ for 20 min, BF₃·Et₂O (3.90 mmol, 1.1 mL) was added via a syringe. After 2 h, TEA (7.80 mmol, 1.0 mL) was added to neutralize excessive acid, then DDQ (13.65 mmol, 3.100 g) was added and the resulting mixture was stirred for an additional 1 h. The residues were purified by column chromatography on silica gel eluted with Hex/DCM ($V_{Hex}:V_{DCM} = 4:1$). Both yields for F₁₂TPP and F₂₀TPP are 24%. F₁₂TPP and F₂₀TPP were all evidenced by [1]H NMR, [19]F NMR (Supplementary Figs. 69–72) and elemental analysis. F₁₂TPP: analysis (calcd., found for $C_{44}H_{18}F_{12}N_4 \cdot 0.3H_2O$): C (63.21, 63.49), H (2.24, 2.59), N (6.70, 6.77). F₂₀TPP: analysis (calcd., found for $C_{44}H_{10}F_{20}N_4 \cdot 0.7H_2O \cdot 0.1C_6H_{14}$ (0.1 Hex)): C (53.80, 53.85), H (1.30, 1.31), N (5.63, 5.81).

### Preparation of FₓChs

FₓChs were synthesized based on modified methods in the literature[48,49]. Note: the chlorin-based compounds F₀Ch[48], F₂₀Ch[71], and F₂₀BC[72] have been previously reported. F₄Ch and F₁₂Ch are new compounds.

For F₀Ch, F₄Ch, and F₁₂Ch: The corresponding porphyrin (0.50 mmol), p-toluenesulfonylhydrazine (TSH, 2.00 mmol, 373 mg), and anhydrous K₂CO₃ (5.00 mmol, 691 mg) were dissolved in dry pyridine (50 mL). The mixture was heated at 105 °C under N₂ in the dark for 12 h, during which TSH (2.00 mmol, 373 mg) was added every 3 h. For F₂₀Ch: F₂₀TPP (0.51 mmol, 497 mg), TSH (2.55 mmol, 475 mg), and anhydrous K₂CO₃ (2.75 mmol, 380 mg) were dissolved in dry pyridine (50 mL) and the mixture was heated at 105 °C for 4 h under N₂ in the dark.

Purification procedure: After cooling to room temperature, the reaction mixture was added to 200 mL water and then extracted with DCM. The extracted organic portion was washed with 2 M HCl (3 times), saturated sodium bicarbonate aqueous solution (2 times) and deionized water (3 times). Appropriate amounts 2,3-dichloro-5,6-dicyano-benzoquinone (DDQ, 1 mg/mL) in DCM were slowly added to the collected DCM layer until the characteristic absorption of the over-reduced product at -740 nm (for synthesis of F₀Ch and F₄Ch), -744 nm (for synthesis of F₁₂Ch), -748 nm (for synthesis of F₂₀Ch) disappeared. The solvent was removed and the crude product was purified by silica gel column chromatography using DCM (for purification of F₀Ch, F₄Ch, and F₁₂Ch) or PE/DCM ($V_{PE}:V_{DCM} = 50:1$) (for purification of F₂₀Ch) as the eluent to give the corresponding chlorin, which were characterized by [1]H NMR, and/or [13]C NMR and/or [19]F NMR spectra, C/H/N elemental analysis, and HRMS.

F₀Ch: (yield: 57%), [1]H NMR (400 MHz, CDCl₃) δ 8.56 (d, J = 4.9 Hz, 2H), 8.41 (s, 2H), 8.17 (d, J = 4.9 Hz, 2H), 8.10 (dd, J = 7.5, 1.7 Hz, 4H), 7.93–7.84 (m, 4H), 7.67 (dt, J = 8.8, 4.5 Hz, 12H), 4.16 (s, 4H), −1.43 (s, 2H); HRMS (m/z): $[M + H]^+$ calcd. for $[C_{44}H_{33}N_4]^+$, 617.26997; found, 617.26898; analysis (calcd., found for $C_{44}H_{32}N_4 \cdot 0.4H_2O$): C (84.70, 84.77), H (5.30, 5.47), N (8.98, 8.96).

F₄Ch: (yield: 17%), [1]H NMR (400 MHz, CDCl₃) δ 8.57 (d, J = 4.9 Hz, 2H), 8.40 (s, 2H), 8.22 (d, J = 4.9 Hz, 2H), 8.10 −7.95 (m, 2H), 7.83 (dt, J = 15.6, 7.8 Hz, 2H), 7 7.70 (m, 4H), 7.46 (m, 8H), 4.28–4.13 (m, 4H), −1.44 (s, 2H); [13]C NMR (151 MHz, CDCl₃) δ 168.10, 162.63 (m), 161.00 (m), 152.58, 140.58, 135.75 (m), 135.25, 134.82 (m), 131.95, 130.35 (d, J = 7.7 Hz), 130.21 (d, J = 8.2 Hz), 130.11 (d, J = 17.5 Hz), 129.50 (d, J = 16.5 Hz), 127.95, 124.13 (m), 123.36, 122.93 (m), 116.16 (m), 115.55, 115.29 (m), 105.69, 35.38; [19]F NMR (376 MHz, CDCl₃) δ −111.29− −112.26 (m, 2F), −112.67− −113.58 (m, 2F); HRMS (m/z): $[M + H]^+$ calcd. for $[C_{44}H_{29}F_4N_4]^+$, 689.23229; found, 689.23102; analysis (calcd., found for $C_{44}H_{28}F_4N_4 \cdot 0.4H_2O$): C (75.94, 75.67), H (4.17, 4.45), N (8.05, 7.97).

$F_{12}Ch$: (yield: 70%), $^1H$ NMR (400 MHz, $CDCl_3$) δ 8.64 (d, $J = 5.0$ Hz, 2H), 8.45 (s, 2H), 8.30 (d, $J = 4.9$ Hz, 2H), 7.11 (dt, $J = 8.5$, 4.0 Hz, 8H), 4.25 (s, 4H), −1.47 (s, 2H); $^{13}C$ NMR (151 MHz, $CDCl_3$) δ 168.87, 164.18 (m), 163.10 (m), 162.51 (m), 161.44 (m), 152.76, 140.52, 135.40, 131.87, 127.88, 123.24, 115.39 (m), 115.06 (m), 107.97, 101.02 (m), 100.32 (m), 98.27, 35.15; $^{19}F$ NMR (376 MHz, $CDCl_3$) δ −105.52 (d, $J = 6.7$ Hz, 4F), −106.57 (t, $J = 6.5$ Hz, 2F), −106.66−−106.88 (m, 6F); HRMS (m/z): $[M + H]^+$ calcd. for $[C_{44}H_{21}F_{12}N_4]^+$, 833.15691; found, 833.15497; analysis (calcd., found for $C_{44}H_{20}F_{12}N_4$): C (63.47, 63.50), H (2.42, 2.60), N (6.73, 6.77).

$F_{20}Ch$: (yield: 21%), $^1H$ NMR (400 MHz, $CDCl_3$) δ 8.68 (d, $J = 5.0$ Hz, 2H), 8.45 (s, 2H), 8.35 (d, $J = 5.0$ Hz, 2H), 4.31 (s, 4H), −1.53 (s, 2H); $^{19}F$ NMR (376 MHz, $CDCl_3$) δ −136.83−−137.13 (m, 4F), −137.91 (dd, $J = 23.9$, 8.5 Hz, 4F), −152.02 (dt, $J = 85.0$, 20.9 Hz, 4F), −160.69 (m, 4F), −161.60 (m, 4F); HRMS (m/z): $[M + H]^+$ calcd. for $[C_{44}H_{13}F_{20}N_4]^+$, 977.08154; found, 977.07904; analysis (calcd., found for $C_{44}H_{12}F_{20}N_4$): C (54.12, 54.42), H (1.24, 1.54), N (5.74, 5.94).

## Preparation of $F_{20}BC$

$F_{20}BC$ was synthesized based on a modified method in the literature[72]. $F_{20}TPP$ (0.13 mmol, 125 mg,) and TSH (3.90 mmol, 740 mg) were added to a mortar, and then grinded evenly. The powder was put into a Schlenk flask and kept under vacuum for 12 h. Subsequently, the mixture was heated to 160 °C and kept for 30 min. After cooled to room temperature, the mixture was purified by silica gel column chromatography using PE/DCM ($V_{PE}:V_{DCM} = 20:1$) as the eluent and washed with Hex to obtain the corresponding $F_{20}BC$ (8% yield). Recrystallization of $F_{20}BC$ by vapor diffusion of Hex into a chloroform solution gave block green crystals suitable for X-ray diffraction analysis. $F_{20}BC$ were evidenced by $^1H$ NMR, $^{19}F$ NMR (Supplementary Figs. 82 and 83) and elemental analysis (calcd., found for $C_{44}H_{14}F_{20}N_4 \cdot 0.3C_6H_{14}$ (0.3 Hex)): C (54.77, 54.79) H (1.83, 1.91), N (5.58, 5.61).

## Characterization

$^1H$ NMR and $^{19}F$ NMR spectra were recorded on a Bruker advance III 400-MHz NMR instrument at room temperature. UV−vis spectra were acquired using a Thermo Scientific GENESYS 50 UV-visible spectrophotometer. HRMS spectra were collected on a Thermo Scientific Orbitrap Q Exactive ion trap mass spectrometer. Dynamic light scattering experiments were conducted with a Brookhaven Elite Sizer zatapotential and a particle size analyzer. C/H/N analysis for all the photosensitizers, catalyst, and electron donor were recorded on vario EL cube elemental analyzer.

## Photocatalytic $CO_2$ reduction

A typical photocatalytic $CO_2$ reduction experiment was carried out in a glass vial (56.8 mL) upon successive addition of DMF solution (5 mL) containing BIH, FeTDHPP, and the chromophore. The glass vial equipped with a magnetic stirrer was sealed with an airtight rubber plug and purged with $CO_2$ for at least 25 min. The reaction sample was then irradiated with a red LED light setup (λ = 630 nm or 730 nm, PCX-50C, Beijing Perfectlight Technology Co., Ltd.). The gaseous products in the headspace were analyzed by Shimadzu GC-2014 gas chromatography equipped with Shimadzu Molecular Sieve 13X 80/100 3.2 × 2.1 mm × 3.0 m and Porapak N 3.2 × 2.1 mm × 2.0 m columns. A thermal conductivity detector (TCD) was used to detect $H_2$ and a flame ionization detector (FID) with a methanizer was used to detect CO and other hydrocarbons. Nitrogen was the carrier gas. The oven temperature was kept at 60 °C. The TCD detector and injection port were kept at 100 °C and 200 °C, respectively. $^{13}C$ isotopic labeling experiments were conducted in a $^{13}CO_2$ atmosphere and the gas products were analyzed by GC-MS (Thermo Scientific TSQ Quantum XLS).

## Photolysis quenching

During photolysis, 2.52 μmol $Co(dmgH)_2PyCl$ in DMF (210 μL) was injected into a photocatalytic solution under $N_2$ and the mixture was allowed to stir for 3 h in the dark. The mixture was then exposed to the air for 1 h and analyzed by UV−vis spectroscopy. Direct exposure of the reaction mixture to the air led to complicated oxidized species with unidentified UV−vis spectra.

## Electrochemical measurements

Cyclic voltammetry (CV) and square wave voltammetry (SWV) were performed on a CHI-760E electrochemical workstation, using a glassy carbon working electrode (diameter 3 mm), Pt auxiliary electrode, and a SCE reference electrode. The electrolyte was 0.1 M tetrabutylammonium hexafluorophosphate ($TBAPF_6$) in DMF or $DMF/H_2O$. The solution was purged with $N_2$ or $CO_2$ at least 20 min before measurements. All reported potentials in this work are versus SCE.

## Fluorescence and excited-state lifetime determination

A solution of the chromophore in a closed quartz cuvette with a septum cap was purged with $N_2$ for at least 15 min. The steady-state fluorescence was recorded on the Duetta fluorescence and absorbance spectrometer. The excited-state lifetimes ($\tau_0$) of $F_xCh$ were measured with an FLS 980 or FLS 1000 fluorescence spectrometer (Edinburgh instruments), in which a picosecond pulsed diode laser (λ = 472 nm) (Edinburgh instruments EPL470) was used as the excitation source.

## Quantum yield measurement

The experiments were conducted on 630 nm LED light. The blank was a DMF solution containing 5 μM FeTDHPP, and 50 mM BIH. The difference between the power (P) of light passing through the blank and through the sample containing $F_xCh$ (x = 0, 4, 12, 20) was measured with a FZ-A Power meter (Beijing Normal University Optical Instrument Company). The quantum yield ($\Phi$) was calculated according to the Eq. (3):

$$\Phi_{CO} = \frac{\text{number of CO molecules}}{\text{number of incident photons}} \times 100\% \quad (3)$$

that is,

$$\Phi_{CO} = \frac{n(CO)}{I} \times 100\% \quad (4)$$

where $n(CO)$ is the number of molecules of CO produced, $I$ is the number of incident photons; $I$ can be calculated by the Eq. (5):

$$I = PSt\frac{\lambda}{hc} \times 100\% \quad (5)$$

$S$ is the incident irradiation area ($S = 6.33$ cm$^2$), $t$ is the irradiation time (in second), $\lambda$ is the wavelength of the light (630 nm), $h$ is the Plank constant ($6.626 \times 10^{-34}$ J·s), and $c$ is the speed of light propagation ($3 \times 10^8$ m·s$^{-1}$).

$$\Phi_{CO} = \frac{n \times N_A}{PSt \times \frac{\lambda}{hc}} \times 100\% \quad (6)$$

where $N_A$ is the Avogadro constant ($6.02 \times 10^{23}$ mol$^{-1}$).

## X-ray crystallography

X-ray diffraction data were collected on SuperNova single crystal diffractometer using the CuKα (1.54184 nm) radiation at 150 K. Absorption correction was carried out by a multiscan method. The crystal structure was solved by direct methods with SHELXT[73] program, and

was refined by full-matrix least-square methods with SHELXL[73] program contained in the Olex2-1.5[74]. Weighted R factor ($Rw$) and the goodness of fit $S$ were based on $F_2$, conventional R factor ($R$ was based on $F$ (Supplementary Table 9). Hydrogen atoms were placed with the AFIX instructions and were refined using a riding mode. Figures were drawn with Diamond software.

## Data availability

The data that support the findings of this study are available from the corresponding author on request. The X-ray crystallographic data for $F_{20}BC$ reported in this study have been deposited at the Cambridge Crystallographic Data Centre (CCDC), under deposition numbers 2289797. These data can be obtained free of charge from The Cambridge Crystallographic Data Centre via www.ccdc.cam.ac.uk/data_request/cif. All other data generated in this study are provided in the Supplementary Information/Source data file. Source data are provided with this paper.

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

## Acknowledgements

We are grateful for the financial support provided by Sun Yat-sen University, GBRCE for Functional Molecular Engineering, and the China Postdoctoral Science Foundation (2022TQ0380 H.Y.; 2022M723586 H.Y.). We thank Z. Yang for providing instrumental support for fluorescence measurements.

## Author contributions

Z.H. conceived the research. Z.H. and S.Y. designed the experiments and wrote the manuscript. S.Y. performed synthesis of chromophores, light-driven, and UV–vis experiments. H.Y. synthesized FeTDHPP and performed DLS tests. K.G. and L.J. collected and analyzed the crystallographic data. S.Y., Z.W., and M.M. performed electrochemical measurements. J.Y. assisted in photolysis experiments using $F_4$TPP. All authors analyzed data.

## Competing interests

The authors declare no competing interests.
