## [Peer Review File · Nature Communications]

Fluorinated chlorin chromophores for red-light-driven CO₂ reductionReviewers' Comments:

Reviewer #1:

Remarks to the Author:

This manuscript presents a family of fluorinated chlorins tested as photosensitizers for the red-light-driven photocatalytic conversion of CO₂ to CO in a multi component homogeneous system containing a Fe porphyrin catalyst and BIH as electron donor, in DMF. Thus, the authors present a system which is based only on earth-abundant elements. The system is also robust (e.g., it lasts over 240 hours) and stays active under 1% concentration of CO₂. The notable aspect of the current study is the improved efficiency of the system (homogeneous), in comparison with those already reported (homogeneous and heterogeneous). The results of this work are of interest in the artificial photosynthesis community and to the larger research community as they open an effective approach for the rational design of efficient chromophores for low-energy light conversion systems.

The work is well done, the experiments have been well performed with a range of techniques being employed, including photophysical and electrochemical analyses. A substantial amount of data is presented and the paper is well written, with care to details. This is a very nice study. The isotopic labeling experiments are convincing, together with all the other control experiments. The conclusions are well supported by the data presented. I recommend publication after the following points are addressed.

- The authors mention that CO is the major product. Were there other products detected in the gas phase (e.g., hydrogen, methane)? Were there other products detected in the liquid phase (e.g., formate/formic acid, methanol)? There is no mention on how the liquid phase was analysed. The selectivity of the system for CO is not given.

- As the accuracy of the concentrations reported in this type of studies for the photosensitizers, the catalysts and the electron donors is paramount, the purity of the solids in bulk for each of these compounds is very important. Therefore, under these circumstances, it is important to have confirmation of purity for the solids in bulk by C/H/N elemental analysis. Characterization techniques such as NMR and HRMS are indicative of the identity and to a certain extent of the purity of compounds. They cannot exclude, though, the presence of small amounts of inorganic salts, for example.

- Were mercury poisoning experiments performed? They would confirm the homogeneous nature of the photocatalysis in the system under study.

- Based on literature reports and/or TD-DFT studies, could the authors propose the assignment of the UV-vis spectra bands for the species under study with respect to the type of electronic transitions involved?

- Regarding electrochemical data, were the experiments performed under N₂ + 1% H₂O, as well?

- It will be interesting to see the profile of the TOF (at least in Supplementary Information) for the experiments shown in Fig. 2 b, c, d, e.

- There is no discussion on the XRD structure – it would be of interest to see (at least in Supplementary Information) a comparison in terms of bond lengths, supporting the reduced nature of this species. In addition, a CCDC search reveals that there are several reports for the solid state structures of chlorins (e.g., JIEMIL, QGDAZ, GELGUZ, GELJEM, GELQAP, SAZROC, BAXWOQ, VIQWAU) and bacteriochlorophylls (e.g., FUBJIV, SAZRES, SAZRIW, SAZRUI). Comparison in terms of bond lengths with these structures would be of interest as well.

- The authors state (lines 77-79 in the manuscript) : *'The TON described here is significantly higher*

than those reported for red-light-driven CO₂RR systems including the ones using noble metals (Supplementary Tables 1-2).'. It would be pertinent to consider stating: 'The TON described here is significantly higher than those reported for red-light-driven CO₂RR (**homogeneous and heterogeneous**) systems including the ones using noble metals (Supplementary Tables 1-2).'

- In the Supplementary Information Table 1, it will be of interest to add a column with the solvent/sacrificial donor system and to the column TON, to add the time in parenthesis, transforming it in TON (time). In the Supplementary Information Table 2, it will be of interest to also add the solvent, maybe to the column Electron donor, transforming it in Electron donor/solvent.

- Could the chlorin-based photosensitizer and/or intermediates and the catalyst also have favourable electrostatic interactions? Could this play a role in the enhanced performance?

Minor points:

- As a suggestion, the reader would benefit by also finding in Fig. 1 the structures of the catalyst and the BIH.

- It is not clear if the chlorin-based compounds reported herein are new compounds or they were already reported.

- Fig. 4 doesn't really present the mechanism of the CO₂RR. It presents the intermediates in the photosensitizer cycle. Therefore, please change the title of Fig. 4 accordingly.

- Line 77 in the manuscript – the units for TOF should be h⁻¹.

- Top of page S3 – please review: '... the reaction mixture was added with 200 mL water ...'.

Reviewer #2:

Remarks to the Author:

Han and coworkers reported a red-light-responsive photosensitizer, fluorinated chlorin, for homogenous photocatalytic CO₂ reduction. The significance of this study is low because the catalytic performance might have been exaggerated (see details below) and many important metrics of photocatalytic CO₂ reduction were not provided. Therefore, I don't recommend its publication in Nature Communications. My specific comments are as follows.

1. It seems that the catalyst and photosensitizer were decomposed in the catalytic process, and huge amounts of them were required to keep the claimed "stability". Also, in line 8 from the bottom of page 7, the authors admitted that FeTDHPP was completely decomposed. If they are consumed, they cannot even be named catalyst and photosensitizer. Therefore, the catalytic performance is exaggerated, and the significance of this study is low.

2. Only CO production and TON were presented. The selectivity, CO₂ conversion, quantum efficiency, and solar-to-chemical efficiency should be provided as well.

3. All the products should be analyzed. It seems that the liquid was not analyzed at all. HCOOH and other chemicals might be produced.

4. What's the light intensity?

5. Control experiments (activity tests, GC-MS measurements, CV analysis, etc.) were performed under N₂. N₂ is not really inert. Ar or He should be employed instead.

6. A scheme illustrating the entire CO₂ reduction process (cycle) should be provided as a replacement of Fig. 3.

Reviewer #3:

Remarks to the Author:

This study by Han and co-workers is a very interesting body of work. While the overall TONs are impressive at first glance the large excess of BIH used and the limited mechanistic studies to truly understand the PCET chemistry occurring in this system are found lacking. The content is well written and the science presented is of reasonable quality however there is still many questions to be answered before publication in Nature, mostly concerning the role of BIH wrt PCET mechanisms with both photosensitizer and catalyst.

Major revisions:

I am somewhat surprised by the authors photocatalysis reaction conditions, e.g. using 5 mL DMF, 50mM BIH (5x10⁴ equivalents SED wrt catalyst), 1 μ M FeTDHPP catalyst, and 50 μ M photosensitizer. Did the authors not add any base, e.g. TEA, to their DMF solvent? The chemistry of BIH as a SED is fairly well established now. Single electron oxidation generates the BIH-radical cation. It is necessary to deprotonate this intermediate (typically with TEA) to make the BI-radical available for a second DARK electron transfer step to fully activate the catalyst. Thus, upon single photon absorption BIH donates a total of two-electrons (important for accurate QY calculations) and the TEAH-cation may provide a proton for C-OH bond cleavage at the catalyst for CO evolution. So not adding any TEA or alternative base to the catalysis system restricts this chemistry to the BIH-radical cation intermediate. This chemistry is further complicated by the fact that the Fe(III) porphyrin catalyst requires a 3e activation to access its Fe(0) active state, thus consuming 3e for initially activation and 2e for each propagated catalytic cycle once activated. At high TONs, the initial 3e activation can likely be neglected. Interestingly, where the authors report QY eqns. they assume (not explicitly) that each cycle requires just a single photon (i.e. the numerator in eq.1 page SI-5 is not multiplied by 2). This therefore assumes that their BIH is donating 2e per photon. The 2e donating properties of BIH are only established in the presence of TEA to generate the deprotonated BI-radical intermediate; unless there is a H-atom transfer step from the BIH-radical cation. This opens upon the question of H-atom transfer from the BIH-radical cation to the activated catalyst to potentially generate a metal-hydride intermediate (this requires appropriate BDFEs for exergonic HAT chemistry). However, such HAT chemistry initiated by the BIH-radical cation generally results in selective CO₂ insertion at the M-H bond and formate as the primary product of 2e induced CO₂ reduction. Thus I am at a loss as to the role of BIH in the authors catalytic system wrt propagation of the FDHTPP catalyst.

The authors mechanistic studies do suggest that the BIH-radical cation is reacting with their chlorins photosensitizers, via visible absorption evidence of 2e-/2H⁺ reduced chlorinphlorins. Yet, as discussed above, BIH is known to be just a 2E-/1H⁺ donor, so just a single proton donor, and even then only when in the presence of a strong base such as TEA. Keeping in mind that residual H₂O in an insufficient base for BIH-radical cation deprotonation in DMF, can the authors address the lack of stoichiometry here and how such chemistry may occur in the absence of any added base? While I appreciate the authors focus is on their photosensitizers, even though there are still question here wrt the PCET chemistry of BIH (and oxidized versions thereof) with chlorins, I believe this is the first report of photocatalytic CO₂ reduction by the FDHTPP catalyst with BIH and this chemistry which is very critical to this study is not reported (to the best of my knowledge) nor is it addressed in any shape or form by the authors here.

The very large excess of BIH used (4 orders of magnitude excess) is also never addressed. At least when utilized in the presence of TEA, BIH is often the limiting reagent wrt TON. Although there are some high TONs here report wrt related literature the relative ratios of BIH are not taken into account here. In this respect the TONs are actually relatively low. I strongly suggest that the authors provide some clarity here, at least for comparison to the bulk literature, by providing reference experiments with photocatalysis conducted in DMF:TEA 4:1 this would very much benefit the readers. Note, TEOA is to be avoided as it is not basic enough to fully deprotonate the BIH-radical cation (see Fujita and co-workers J. Am. Chem. Soc. 2020, 142, 2413–2428)

Minor:

Abstract "However, a molecular system that mimics such function has not been demonstrated in non-noble-metal catalysis" This is not true as per the article by Julia Weinstein and co-workers, *Inorg. Chem.* 2022, 61, 34, 13281–13292. The authors cite this article further below (#21). Although its TON is low it still is fact and makes the authors claim here false.

P2 L21-23 "Because of the large energy barriers associated with the activation of CO₂, high energetic light is generally required in driving such transformation."

This statement is very crude and represents a poor interpretation of the recent literature wrt the general photocatalytic CO₂ reduction literature. Whether using noble or non-noble metals there is a growing literature using red light for photocatalytic CO₂ reduction, as the authors have pointed out and cited in their following paragraph. Therefore they are contradicting themselves in stating that high energy light is a requirement. Recent literature they have cited demonstrates that this is not the case. High energy light is not required – it has simply been the most commonly utilized region of the spectrum in photocatalytic CO₂RR based upon earlier catalytic design; light restrictions of prior research in this field should not be interpreted as fundamental thermodynamic requirements for the activation of CO₂ toward CO production. The growing literature of red-light induced CO₂ reduction photocatalysis simply demonstrates a poor design of previous photocatalytic systems reported in the literature-to date which are not capable of harvesting a lower energy red-light input for CO₂ reduction.

Reviewer #4:

Remarks to the Author:

The manuscript entitled "Fluorinated chlorin chromophores for red-light-driven CO₂ reduction" is an interesting paper proceeding in the presence of an iron based complex FeTDHPP (that is in precious-metal free conditions).

My main question concerns the precautions taken by the authors to ensure that the system is not contaminated by impurities of precious metals (but also copper, nickel...), which could possibly be wholly or partly responsible for the CO₂ reduction.

For example, what is the commercial purity (transition metal based) of the iron salts used in the synthesis of FeTDHPP (98.6.. or 99.99..) ? I did not find the information neither in the paper nor in the ref 2 indicated for its synthesis.

I have the same question regarding the purity of the K₂CO₃ used to prepare FxChs from FxTPP. What is its commercial purity? (It's not uncommon for traces of transition metals to be present in such a base). Even if in this case I think there's no problem because the TON of F12TPP is more or less comparable to that of F12CH (Fig 2a), having the information would be interesting.

Also, I understand that the commercial solvent (DMF) is not purified/distillated before using. Is it the case?

Overall, I don't think there's any impurity problem, but at least one test under perfect conditions with extremely pure reagents (including FeTDHPP), would be more reassuring and would reinforce these interesting results.

Reviewer #1:

This manuscript presents a family of fluorinated chlorins tested as photosensitizers for the red-light-driven photocatalytic conversion of CO₂ to CO in a multi component homogeneous system containing a Fe porphyrin catalyst and BIH as electron donor, in DMF. Thus, the authors present a system which is based only on earth-abundant elements. The system is also robust (e.g., it lasts over 240 hours) and stays active under 1% concentration of CO₂. The notable aspect of the current study is the improved efficiency of the system (homogeneous), in comparison with those already reported (homogeneous and heterogeneous). The results of this work are of interest in the artificial photosynthesis community and to the larger research community as they open an effective approach for the rational design of efficient chromophores for low-energy light conversion systems.

The work is well done, the experiments have been well performed with a range of techniques being employed, including photophysical and electrochemical analyses. A substantial amount of data is presented and the paper is well written, with care to details. This is a very nice study. The isotopic labeling experiments are convincing, together with all the other control experiments. The conclusions are well supported by the data presented. I recommend publication after the following points are addressed.

We would like to thank Reviewer 1 for his/her insightful reading of our manuscript and for the helpful comments and suggestions. Each point is addressed below.

(1) The authors mention that CO is the major product. Were there other products detected in the gas phase (e.g., hydrogen, methane)? Were there other products detected in the liquid phase (e.g., formate/formic acid, methanol)? There is no mention on how the liquid phase was analysed. The selectivity of the system for CO is not given.

We thank the referee for the comment. We regret that we did not include the selectivity of CO in the manuscript. We have investigated this further. There was no other gaseous product detected besides CO from experiments conducted under one atmosphere of CO₂. For experiments performed under low concentrations of CO₂, we did observe small amounts of H₂ from the systems, in which the selectivity was 95.6% (under 1% CO₂) and 97.7 % (under 5% CO₂). Regarding to the analysis of liquid products, we have tried to use ¹H NMR to detect formic acid and methanol. However, none of them could be observed. In this particular experiment, we were able to generate a large amount of CO (232 μmol) from the system. Our control experiments showed that even though the selectivity of formic acid or methanol was as low as 1%, we should be able to detect them by ¹H NMR. Thus, we conclude that there was no liquid product generated from the system.

In addition, we have performed photocatalytic experiments with high concentrations of FeTDHPP and F₂₀Ch. The results showed that the amounts of CO generated were very close to the theoretical yield of electron that can be provided from BIH (based on two

electrons per BIH molecule). Please see the following Figure S13. This evidence along with our GC and NMR measurements strongly suggest that the selectivity of CO is nearly 100%.

We have revised the manuscript by including the following statements to page 5:

“In all the red-light-driven experiments performed under an atmosphere of CO₂, no other gaseous product was observed by GC. Analysis of the liquid phase by ¹H NMR showed no detection of formic acid and methanol. To study the selectivity of the system further, we found that the amounts of CO generated were near the theoretical maximum value (Supplementary Fig. 13) of the electron donor BIH (based on two electrons per BIH molecule), which suggests that the selectivity of CO is nearly 100%.”

“Indeed, we observed slower rates of CO generation from mixtures at lower concentrations of CO₂ (Fig. 2f). However, its ability to function at low CO₂ contents (down to 1%) with high selectivities (97.7% under 5% CO₂ and 95.6% under 1% CO₂) was impressive.”

Supplementary Figure 13. Photocatalytic CO₂ reduction. Time profiles of photocatalytic CO₂ reduction in CO₂-saturated DMF solutions containing 100 μM FeTDHPP, 100 μM F₂₀Ch and 20 mM BIH under red LED ($\lambda = 630$ nm, 110 mW/cm²) at 293 K. Error bars denote standard deviations based on at least three separated runs.

(2) As the accuracy of the concentrations reported in this type of studies for the photosensitizers, the catalysts and the electron donors is paramount, the purity of the solids in bulk for each of these compounds is very important. Therefore, under these circumstances, it is important to have confirmation of purity for the solids in bulk by C/H/N elemental analysis. Characterization techniques such as NMR and HRMS are

indicative of the identity and to a certain extent of the purity of compounds. They cannot exclude, though, the presence of small amounts of inorganic salts, for example.

We thank the referee for the comment. We have performed C/H/N analysis for all the photosensitizers, catalyst, and electron donor in our study. And the results suggest they are pure. Please see the results listed below. We have included these results in the SI.

F₀TPP: Anal. Calcd. For C₄₄H₃₀N₄•0.8H₂O: C, 84.00; H, 5.06; N, 8.91; found: C, 84.00; H, 5.07; N, 8.81.

F₄TPP: Anal. Calcd. For C₄₄H₂₆F₄N₄•0.3H₂O: C, 76.36; H, 3.87; N, 8.10; found: C, 76.49; H, 4.18; N, 8.06.

F₁₂TPP: Anal. Calcd. For C₄₄H₁₈F₁₂N₄•0.3H₂O: C, 63.21; H, 2.24; N, 6.70; found: C, 63.49; H, 2.59; N, 6.77.

F₂₀TPP: Anal. Calcd. For C₄₄H₁₀F₂₀N₄•0.7H₂O•0.1C₆H₁₄ (0.1 Hex): C, 53.80; H, 1.30; N, 5.63; found: C, 53.85; H, 1.31; N, 5.81.

F₀Ch: Anal. Calcd. For C₄₄H₃₂N₄•0.4H₂O: C, 84.70; H, 5.30; N, 8.98; found: C, 84.77; H, 5.47; N, 8.96.

F₄Ch: Anal. Calcd. For C₄₄H₂₈F₄N₄•0.4H₂O: C, 75.94; H, 4.17; N, 8.05; found: C, 75.67; H, 4.45; N, 7.97.

F₁₂Ch: Anal. Calcd. For C₄₄H₂₀F₁₂N₄: C, 63.47; H, 2.42; N, 6.73; found: C, 63.50; H, 2.60; N, 6.77.

F₂₀Ch: Anal. Calcd. For C₄₄H₁₂F₂₀N₄: C, 54.12; H, 1.24; N, 5.74; found: C, 54.42; H, 1.54; N, 5.94.

F₂₀BC: Anal. Calcd. For C₄₄H₁₄F₂₀N₄•0.3C₆H₁₄ (0.3 Hex): C, 54.77; H, 1.83; N, 5.58; found: C, 54.79; H, 1.91; N, 5.61.

FeTDHPP: Anal. Calcd. for C₄₄H₂₈ClFeN₄O₈•1.7H₂O•C₆H₁₄(Hex): C, 63.29; H, 4.82; N, 5.90; found: C, 63.39; H, 5.11; N, 5.70.

BIH: Anal. Calcd. For C₁₅H₁₆N₂: C, 80.32; H, 7.19; N, 12.49; found: C, 80.20; H, 7.26; N, 12.42.

We have also conducted photocatalytic experiments in freshly distilled DMF with chemicals that passed the EA tests. The results were identical to the ones we presented previously. Please see the following Figure:

Photocatalytic CO₂ reduction in CO₂-saturated DMF containing 50 μM F₂₀Ch, 1 μM FeTDHPP, and 50 mM BIH under red LED ($\lambda = 630$ nm, 110 mW/cm²) at 293 K for 51 h. Error bars denote standard deviations, based on at least three separated three runs.

In addition, we have performed several control experiments to rule out potential contaminants from inorganic salts. Please see the following table. Inorganic salts (Fe³⁺, Cu²⁺, Ni²⁺, Co³⁺, Ru³⁺) with or without adding the TDHPP ligand ALL produced no or negligible amount of CO as compared with the experiment using FeTDHPP as the catalyst.

Supplementary Table 7. Control experiments with inorganic salts.

Entry	Catalyst	CO (μmol)	H ₂ (μmol)
1	FeTDHPP (1 μM)	8.2 ± 0.63	0
2	Fe(NO ₃) ₃ (10 μM)	0.1	trace
3	Cu(NO ₃) ₂ (10 μM)	0	0.17
4	Ni(NO ₃) ₂ (10 μM)	0.14	trace
5	Co(NO ₃) ₃ (10 μM)	0	trace
6	RuCl ₃ (10 μM)	0.15	trace
7	Fe(NO ₃) ₃ (1 μM) + TDHPP (1 μM)	0.12	0
8	Cu(NO ₃) ₂ (1 μM) + TDHPP (1 μM)	0.11	0
9	Ni(NO ₃) ₂ (1 μM) + TDHPP (1 μM)	0.13	0
10	Co(NO ₃) ₃ (1 μM) + TDHPP (1 μM)	trace	0
11	RuCl ₃ (1 μM) + TDHPP (1 μM)	0.11	0

Reaction conditions: a 5 mL CO₂-saturated DMF solution containing F₂₀Ch (50 μM), BIH (50 mM) and metal catalyst was irradiated using red LED ($\lambda = 630$ nm, 110 mW/cm²) under a CO₂ atmosphere for 27 h.

We have added the following description to the MS:

“To rule out potential metal contaminants, inorganic salts (Fe³⁺, Cu²⁺, Ni²⁺, Co³⁺, Ru³⁺) with or without TDHPP ligand all produced no or negligible amount of CO as compared with the experiment using FeTDHPP as the catalyst (Supplementary Table 7).”

We have also added the following ^1H NMR for BIH and TDHPP ligand to the SI:

Supplementary Figure 81. ^1H NMR spectrum. ^1H NMR spectrum of BIH in DMSO- d_6 .

Supplementary Figure 82. ^1H NMR spectrum. ^1H NMR spectrum of TDHPP in methanol- d_4 .

(3) Were mercury poisoning experiments performed? They would confirm the homogeneous nature of the photocatalysis in the system under study.

We thank the referee for the question. We have performed the experiment suggested by you. Both experiments with or without metallic Hg^0 exhibited similar activity in CO production. Please see the following Figure S15. We have added this figure to the SI and the following statement to the MS to explain this:

“In addition, experiments conducted in the presence of an excess amount of Hg^0 showed an identical activity profile (Supplementary Fig. 15), indicating no contamination from amalgam-forming metals.”

Supplementary Figure 15. Photocatalytic CO_2 reduction. Photocatalytic CO_2 reduction in the presence and absence of Hg (0.02 mL) in CO_2 -saturated DMF solution containing 10 μM FeTDHPP, 50 μM F_{20}Ch , and 20 mM BIH under red LED ($\lambda = 630$ nm, 110 mW/cm^2) at 293 K.

(4) Based on literature reports and/or TD-DFT studies, could the authors propose the assignment of the UV-vis spectra bands for the species under study with respect to the type of electronic transitions involved?

We thank the referee for the question. We have proposed the peak assignment of the UV-Vis spectra. We have revised the manuscript to include the discussion as follows:

“All chlorins exhibit an intense Soret (B) band (from 405 to 420 nm) and 4 to 5 Q bands (from 504 to 654 nm). As would normally be anticipated for chlorins⁴⁸⁻⁵¹, red-shift and the higher extinction coefficient for the Q band between 650-654 nm than those observed for corresponding F_xTPP ”

References 48-51:

Dyes Pigments **2021**, 185, 108886; *J. Am. Chem. Soc.* **2016**, 138, 12502-12510; *Biorg. Med. Chem.* 2003, 11, 1643-1652; *ACS Catal.* **2017**, 7, 3597-3606.

(5) Regarding electrochemical data, were the experiments performed under $N_2 + 1\%$ H_2O , as well?

We thank the referee for the question. We have performed the electrochemical experiments under Ar in the presence of 1% H_2O . Similar spectra were observed with or without 1% H_2O . Please see the following Figure S19. We have added this figure to the SI and included the following statements to the MS:

“Similar CV spectra were obtained in experiments conducted in the presence of 1% H_2O (Supplementary Fig. 19).”

Supplementary Figure 19. Electrochemical study. Cyclic voltammograms of 0.25 mM F_0Ch (black), 0.5 mM F_4Ch (green), 0.5 mM $F_{12}Ch$ (blue), and 0.5 mM $F_{20}Ch$ (red) in DMF containing 0.1 M TBAPF₆ in the presence (solid) and absence (short dot) of 1% H_2O under Ar at a scan rate of $0.1 V \cdot s^{-1}$

(6) It will be interesting to see the profile of the TOF (at least in Supplementary Information) for the experiments shown in Fig. 2 b, c, d, e.

We thank the referee for the suggestion. We have added profiles of TOF for the experiments shown in Fig. 2 b-e to the SI, and have included the following sentence to the caption of Fig. 2 as follows:

“The profiles of TOF for **b-e** were shown in Supplementary Fig. 9, Table 1, and Supplementary Tables 4-5.”

Supplementary Figure 9. Photocatalytic CO₂ reduction. a, TOF_{CO} for systems with different F_xTPP. b, TOF_{CO} for systems with different F_xCh. c, TOF_{CO} for systems with different initial [F₂₀Ch] and [FeTDHPP]. d, TOF_{CO} for stability of a system with F₂₀Ch. Catalytic conditions: (a-b) used 50 μM F_xTPP or F_xCh, 1.0 μM FeTDHPP, and 50 mM BIH; (c) used the same concentrations (1, 2, 5, 10 μM) of F₂₀Ch and FeTDHPP, 50 mM BIH; (d) used 100 μM F₂₀Ch, 100 μM FeTDHPP, and 200 mM BIH. Experiments were in CO₂-saturated DMF (5.0 mL) at 20 °C using a light-emitting diode (LED) source ($\lambda = 630$ nm, 110 mW/cm²).

The TOF profiles of Fig. 2c have been included in Table 1:

Table 1| Photophysical, electrochemical, and photocatalytic CO₂ reduction data of F_xCh

	λ (nm) ϵ ($\times 10^4$ M ⁻¹ cm ⁻¹)	E_{red}^a (V vs. SCE)	E_{red}^b (V vs. SCE)	TON ^c (630 nm)	TOF ^c (h ⁻¹)	TON(F _x Ch) ^c /TON(F _x TPP) ^d	TON ^c (730 nm)
F ₀ Ch	650 (3.56)	-1.11, -1.58	-1.05, -1.81	210 ± 42	8	52.5	36
F ₄ Ch	651 (3.46)	-1.01, -1.52	-0.98, -1.82	558 ± 16	112	3.7	64
F ₁₂ Ch	653 (4.64)	-0.85, -1.40	-0.86, -1.33, -1.78	740 ± 78	26 ± 2	1.4	68
F ₂₀ Ch	654 (5.13)	-0.73, -1.24	-0.72, -1.15, -1.61	1790 ± 52	194 ± 9	1.0	510

Error bars denote standard deviations, based on at least three separated runs. TON and TOF calculated based on [FeTDHPP].

The TOF profiles of Figures 2d and 2e, have been added to Supplementary Table 4:

Supplementary Table 4. Data for photocatalytic CO₂ reduction. Systems containing the same concentration of F₂₀Ch and FeTDHPP with 50 mM BIH in CO₂-saturated DMF under irradiation with red LEDs ($\lambda = 630$ nm, 110 mW/cm²) for 147 h at 293 K. a, 200 mM BIH, amount of CO and TON data collected for 242 h.

[F ₂₀ Chlorin] = [FeTDHPP] (μ M)	CO (μ mol)	TON (CO)	TOF (h ⁻¹)
1	2.1	420	31.8
2	10.1	1010	26.7
5	25.7	1028	16.7
10	70.2	1404	18.8
100 ^a	608.1 \pm 73.1	1216 \pm 146	11.5 \pm 1.5

The TOF profiles of Fig. 2b have been added to Supplementary Table 5.

Supplementary Table 5. Data for photocatalytic CO₂ reduction. Systems containing the 50 μ M F_xTPP (x = 0, 4, 12, 20) and 1 μ M FeTDHPP and 50 mM BIH in CO₂ saturated DMF under irradiation with red LEDs ($\lambda = 630$ nm, 110 mW/cm²) for 75 h at 293 K.

chromophores	CO (μ mol)	TON (CO)	TOF (h ⁻¹)
F ₀ TPP	0.02	4	0.1
F ₄ TPP	0.75	150	12.7
F ₁₂ TPP	2.70	540	10.2
F ₂₀ TPP	8.70	1740	220.7

(7) There is no discussion on the XRD structure – it would be of interest to see (at least in Supplementary Information) a comparison in terms of bond lengths, supporting the reduced nature of this species. In addition, a CCDC search reveals that there are several reports for the solid state structures of chlorins (e.g., JITMIL, QGDAZ, GELGUZ, GELJEM, GELQAP, SAZROC, BAXWOQ, VIQWAU) and bacteriochlorophylls (e.g., FUBJIV, SAZRES, SAZRIW, SAZRUI). Comparison in term of bond lengths with these structures would be of interest as well.

We thank the referee for the comment. We have added discussion for the structure of F₂₀BC in both MS and SI to support the successful synthesis of the complex.

To MS:

“The crystal structure of F₂₀BC shows two characteristic C–C single bonds in the pyrrole ring, which exhibit similar distances compared with reported Ch and BC complexes (Supplementary Table 10)^{57,58}.”

To SI:

Supplementary Table 10. Comparison of bond lengths. The C-C and C=C bond lengths in the pyrrole ring for F₂₀BC and complexes in the literature. The C₇-C₈ bond distance of 1.504(3) Å in F₂₀BC is characteristic of a single bond, which is consistent with previous reported chlorins²⁴ and bacteriochlorins²⁵.

CSD Refcodes of reported Chlorins	Distance of C-C	Distance of C=C	Distance of C=C	Distance of C=C
JITMIL	1.521(9)	1.356(8)	1.368(9)	1.360(9)
GELGUZ	1.525	1.368	1.348	1.361
GELJEM	1.519	1.359	1.363	1.357
GELQAP	1.548	1.366	1.350	1.345
SAZROC	1.545(2)	1.366(2)	1.346(2)	1.364(2)
BAXWOQ	1.5380(16)	1.3651(18)	1.3508(16)	1.36029(17)
VIQWAU	1.539(2)	1.353(2)	1.352(3)	1.339(3)
CSD Refcodes of reported bacteriochlorins	Distance of C-C	Distance of C-C	Distance of C=C	Distance of C=C
F ₂₀ BC (CCDC: 2289797)	1.504(3)	1.504(3)	1.387(3)	1.387(3)
FUBJIV	1.536(4)	1.497(5)	1.366(5)	1.364(4)
SAZRES	1.5365(18)	1.5365(18)	1.3663(18)	1.3663(18)
SAZRIW	1.540(7)	1.532(7)	1.377(7)	1.360(7)
SAZRUI	1.542(3)	1.542(3)	1.373(3)	1.373(3)

(8) The authors state (lines 77-79 in the manuscript): ‘The TON described here is significantly higher than those reported for red-light-driven CO₂RR systems including the ones using noble metals (Supplementary Tables 1-2).’. It would be pertinent to consider stating: ‘The TON described here is significantly higher than those reported for red-light-driven CO₂RR (**homogeneous and heterogeneous**) systems including the ones using noble metals (Supplementary Tables 1-2).’

We thank the referee for the suggestion. The sentence has been corrected according to your suggestion.

(9) In the Supplementary Information Table 1, it will be of interest to add a column with the solvent/sacrificial donor system and to the column TON, to add the time in parenthesis, transforming it in TON (time). In the Supplementary Information Table 2, it will be of interest to also add the solvent, maybe to the column Electron donor, transforming it in Electron donor/solvent.

We thank the referee for the suggestion. We have revised Supplementary Table 1 and Supplementary Table 2 according to your suggestion:

Supplementary Table 1 Summary of molecular systems for CO₂ reduction using low-energy light in the literature.

System	Major product	Solvent/Sacrificial donor	TON (time)	TOF (h ⁻¹)	Φ _{CO} (%)	Light source (nm)	Ref
F ₂₀ Ch + FeTDHPP	CO	DMF/BIH	1842 (51 h)	187 (1.03×10 ⁵ μmol g ⁻¹ h ⁻¹)	0.91	630	This work
F ₂₀ Ch + FeTDHPP	CO	DMF/BIH	510 (170 h)	8 (4.43×10 ³ μmol g ⁻¹ h ⁻¹)	nr	730	This work
Os + Ru(CO)	HCOOH	DMA / BI(OH)H	81 (40 h)	nr	0.061 (480 nm)	725	13
Os + Ru(CO)	HCOOH	DMA / BI(OH)H	42 (12 h)	nr	nr	> 770	13
[ZnTMPyP]Cl ₄ + Mn(bpy)(CO) ₃ Br	CO	H ₂ O/AA	< 1 (~ 4 h)	< 0.1	2.67	625	14
Os(II)-Re(I)(Cl)	CO	DMF/TEOA=5 /BIH	1138 (20 h) (λ > 620 nm)	3.3 min ⁻¹ (λ > 420 nm)	12 (650 nm)	> 620	15

Supplementary Table 2: Summary of heterogeneous systems for CO₂ reduction using low-energy light in the literature.

System	Electron donor/Solvent	Major product	Light source (nm)	Evolution rates (μmol g ⁻¹ h ⁻¹)	Φ _{CO} (%)	Ref
CoN porous atomic layers	Na ₂ S/H ₂ O	CO	800-2500	14.5	nr	16
B ₁₃ P ₂ + Co(bpy) ₃ ²⁺	TEOA/DMF	CO	> 780	6.5	0.07 (810 nm)	17
V _S -AgInS ₂	-/H ₂ O	CO	> 780	8.04	0.055 (790 nm)	18
P ₃	BNAH/MeCN: TEOA=5	CO CH ₄	> 600	282.6 293.7	nr	19
CN + Co(bpy) ₃ ²⁺	TEOA/MeCN: TEOA=3:2	CO	660 730	nr	nr	20
BiOI	-/H ₂ O	CO CH ₄	≥ 700	0.119 μmol h ⁻¹ 0.021 μmol h ⁻¹	0.02 (CO+CH ₄) (700 nm)	21
m-NiAl-LDH	TEOA/CH ₃ CN/TEOA /H ₂ O=3:1:1	CH ₄	> 600	77.2	0.95 (CO+CH ₄) (600 nm)	22
TNP-MOF	TEOA/MeCN	HCOOH	λ ≥ 730	6630 ± 242	2.03 (760 nm) 1.11 (808 nm)	23

(10) Could the chlorin-based photosensitizer and/or intermediates and the catalyst also have favourable electrostatic interactions? Could this play a role in the enhanced performance?

We thank the referee for the question. We have investigated this with UV-vis experiments. Please see the following Figure S53. The results showed that the spectrum of a mixture of photosensitizer and catalyst is simply the sum spectra of photosensitizer and catalyst. Thus, there is no electrostatic interaction between the chlorin and the FeTDHPP. We have added this figure to the SI and included the following statement to the main text:

“No electrostatic interaction was found between the chlorin and FeTDHPP by UV-vis studies (Supplementary Fig. 53), which suggests electron transfer from the chromophore to the catalyst follows an outer-sphere mechanism.”

Supplementary Figure 53: UV-vis absorption spectra. Systems containing 10 μM chromophores (black), 5 μM FeTDHPP (red), a mixture of 10 μM chromophores and 5 μM FeTDHPP, (green) and spectrum of chromophores + spectrum of FeTDHPP (blue). chromophores: F_0Ch (a); F_4Ch (b); F_{12}Ch (c); F_{20}Ch (d).

Minor points:

(11) As a suggestion, the reader would benefit by also finding in Fig. 1 the structures of the catalyst and the BIH.

We thank the referee for the suggestion. We have revised Fig. 1 by adding the structures of FeTDHPP and BIH.

Fig. 1 | Structure diagram. Structures of chlorophyll *a* in nature, chromophores F_xTPP, F_xCh (x = 0, 4, 12, 20), and F₂₀BC, catalyst FeTDHPP, and electron donor BIH in the study.

(12) It is not clear if the chlorin-based compounds reported herein are new compounds or they were already reported.

We thank the referee for the comment. The chlorin-based compounds F₀Ch, F₂₀Ch and F₂₀BC have been previously reported. F₄Ch and F₁₂Ch are new compounds. We have cited relevant papers for the known compounds and have clarified this in the SI:

“Note: the chlorin-based compounds F₀Ch⁸, F₂₀Ch¹⁰ and F₂₀BC⁷ have been previously reported. F₄Ch and F₁₂Ch are new compounds.”

(13) Fig. 4 doesn't really present the mechanism of the CO₂RR. It presents the intermediates in the photosensitizer cycle. Therefore, please change the title of Fig. 4 accordingly.

We thank the referee for the suggestion. We have performed further experiments to come up with a more complete mechanistic scheme for CO₂RR. Please see the following paragraphs and figures added to the manuscript and SI:

“Previous studies showed that BIH functioned as a $2e^-/1H^+$ donor^{59,60}. To generate the BI-radical (which donates the second e^-), deprotonation of the BIH-radical cation by a base such as triethylamine (TEA) was found to be necessary in ACN (Supplementary Table 11). However, there are several photocatalytic studies reported in DMF without TEA or additional bases when using BIH as the electron donor^{53,61,62}. To investigate this, we performed CV studies for BIH in DMF and ACN (Supplementary Figs. 46-47). In contrast to the spectrum in ACN, the CV in DMF showed appearance of a reduction wave at ~ -1.6 V vs SCE, corresponding to generation of the BI-radical. This result suggests that deprotonation of the BIH-radical cation is more favorable in DMF than in ACN. However, addition of TEA to the system was found to improve the activity (Supplementary Fig. 48 and Supplementary Table 11), exhibiting a TON_{CO} of 2132 in 27 h and an initial TOF_{CO} of $584 h^{-1}$. In the overall reactions, BI^+ and OH^- were produced in generation of the $2e^-/2H^+$ reduced chromophores and in CO_2 reduction (Eq. 1 and 2).

The photocatalytic CO_2 reduction mechanism by FeTDHPP has been extensively investigated by Robert and co-workers⁶³⁻⁶⁵. UV-vis studies showed generation of the corresponding Fe(II) and Fe(I) species at the early stage of photolysis (Supplementary Figs. 49-52). The Fe(I) species was found to decrease during CO production, which indicates a catalytic cycle consistent with previous reports (Fig. 4)^{52,64,65}. No electrostatic interaction was found between the chlorin and FeTDHPP by UV-vis studies (Supplementary Fig. 53), which suggests electron transfer from the chromophore to the catalyst follows an outer-sphere mechanism.”

Fig. 4 | Proposed mechanism for the red-light-driven CO_2RR .

Supplementary Figure 46. Electrochemical study. Cyclic voltammograms of 5 mM BIH in MeCN (left) or DMF (right) containing 0.1 M TBAPF₆ under Ar at a scan rate of 0.1 V·s⁻¹

Supplementary Figure 47. Electrochemical study. Cyclic voltammograms of 5 mM BIH in MeCN (bottom) or DMF (top) containing 0.1 M TBAPF₆ under Ar at a scan rate of 0.1 V·s⁻¹

Supplementary Figure 48. Photocatalytic CO₂ reduction. Time profiles of photocatalytic CO₂ reduction in CO₂-saturated DMF (black) or DMF:TEA = 4 :1 (red) solutions containing 50 μM F₂₀Ch, 1 μM FeTDHPP and 50 mM BIH under red LED (λ = 630 nm, 110 mW/cm²) at 293 K.

Supplementary Figure 49: UV-vis absorption spectra. Systems containing 10 μM F_0Ch , 10 μM FeTDHPP and 10 mM BIH in DMF under CO_2 upon irradiation with red LED light ($\lambda = 630 \text{ nm}$, 110 mW/cm^2) in a quartz cuvette (10-mm path length). Irradiation time ranging from 0 to 50 min (a) and from 50 min to 17 h (b).

Supplementary Figure 50: UV-vis absorption spectra. Systems containing 10 μM F_4Ch , 10 μM FeTDHPP and 10 mM BIH in DMF under CO_2 upon irradiation with red LED light ($\lambda = 630 \text{ nm}$, 110 mW/cm^2) in a quartz cuvette (10-mm path length). Irradiation time ranging from 0 to 2 min (a) and from 2 min to 20 min (b) and from 20 min to 13.5 h.

Supplementary Figure 51: UV-vis absorption spectra. Systems containing 10 μM F_{12}Ch , 10 μM FeTDHPP and 10 mM BIH in DMF under CO_2 upon irradiation with red LED light ($\lambda = 630 \text{ nm}$, 110 mW/cm^2) in a quartz cuvette (10-mm path length). Irradiation time ranging from 0 to 5 min (a) and from 5 min to 2 h (b) and from 2 h to 10 h (c).

Supplementary Figure 52: UV-vis absorption spectra. Systems containing 10 μM F_{20}Ch , 10 μM FeTDHPP and 10 mM BIH in DMF under CO_2 upon irradiation with red LED light ($\lambda = 630 \text{ nm}$, 110 mW/cm^2) in a quartz cuvette (10-mm path length). Irradiation time ranging from 0 to 30 min (a) and from 30 min to 2 h (b) and from 2 h to 5 h (c).

(14) Line 77 in the manuscript – the units for TOF should be h^{-1} .

This was corrected.

(15) Top of page S3 – please review: ‘... the reaction mixture was added with 200 mL water ...’.

This sentence has been corrected to ‘... the reaction mixture was added to 200 mL water ...’.

Reviewer #2:

Han and coworkers reported a red-light-responsive photosensitizer, fluorinated chlorin, for homogenous photocatalytic CO_2 reduction. The significance of this study is low because the catalytic performance might have been exaggerated (see details below) and many important metrics of photocatalytic CO_2 reduction were not provided. Therefore, I don't recommend its publication in Nature Communications. My specific comments are as follows.

(1) *It seems that the catalyst and photosensitizer were decomposed in the catalytic process, and huge amounts of them were required to keep the claimed “stability”. Also, in line 8 from the bottom of page 7, the authors admitted that FeTDHPP was completely decomposed. If they are consumed, they cannot even be named catalyst and photosensitizer. Therefore, the catalytic performance is exaggerated, and the significance of this study is low.*

We don't agree with the referee on the comment regarding catalyst deactivation. We have looked into the literature and noticed almost all catalysts and photosensitizers

suffered from deactivation. Catalyst deactivation can be caused by many forms, such as: degradation of the ligand; inhibition by the solvent; inadvertent admission of air or moisture; lability of metal-ligand bonds; inhibition by buildup of reaction products, from inappropriate substrate functionality, from undesired substrate reactions, or by deposition of bulk metal. Turnover number (TON) has been used as one of the most important parameters to evaluate the activity of the catalyst. Please see the following recent examples from the literature:

The first paragraph from “Deactivation in Homogeneous Transition Metal Catalysis: Causes, Avoidance, and Cure” by R.H. Crabtree, in *Chem. Rev.* **2015**, *115*, 127-150:

1. INTRODUCTION

Deactivation leads to loss of catalyst activity or selectivity with increasing reaction time, but has attracted less academic attention in homogeneous catalysis than is justified by its importance. **Ultimate deactivation is inevitable**, but catalyst performance can be greatly affected depending on the balance between the rates of deactivation and of productive catalysis. Even a small improvement in this rate ratio can have a big effect on performance in terms of turnover number (TON) and thus also increases the reaction yield for a given catalyst loading. Only a minority of reports in homogeneous catalysis tackles this problem in detail: not only is it sometimes difficult to identify the deactivation products involved, but workers may also prefer not to spend time studying an apparently failed catalyst or extending the life of a seemingly adequate one. Poater and Cavallo¹ make this point persuasively:

The first paragraph from “Unifying views on catalyst deactivation” by J. Pérez-Ramírez and coworkers, in *Nat. Catal.* **2022**, *5*, 854-866:

Catalysts are not immortal. Their progressive loss of activity or selectivity with time, known as catalyst deactivation, has concerned researchers since their first industrial applications. In his foundational patent on ammonia synthesis, Carl Bosch emphasized that some substances act as poisons and can entirely destroy the catalytic properties of iron¹. Another vivid example is the deactivation of chromium oxide catalysts for the Deacon process, whose activity was said to reduce after one week and thus could not be used for a long period². The lifetime of a catalyst is defined as the reaction time at which regeneration or replacement is necessary, and this depends on its nature, the reaction operating conditions and its performance threshold. It can span from seconds to years for heterogeneous systems³, **whereas the lifespans of homogeneous systems range from minutes to a few hundred hours in continuous mode**⁴, coinciding with typical half-lives of enzymes at their maximum operating temperature⁵. Homogeneous and biocatalysts, however, are typically renewed after each reaction batch, **therefore productivity tends to be more critical than lifetime**^{6,7}.

In fact, FeTDHPP has been extensively studied and identified as an electrocatalyst in previous reports:

Science **2012**, 338, 90-94;

Nature **2017**, 548, 74-77;

J. Am. Chem. Soc. **2014**, 136, 16768-16771;

Coord. Chem. Rev. **2017**, 334, 184-198;

Chemcatchem **2014**, 6, 3200-3207;

Chem. Soc. Rev. **2020**, 49, 5772-5809;

Energy chem. **2020**, 2, 100034.

The deactivation of FeTDHPP has also been studied:

Chemcatchem **2014**, 6, 3200-3207

We have performed further experiments to evaluate the kinetic of the photocatalytic system. The initial rates of CO production indicate a first-order dependence on either the concentration of FeTDHPP or the chlorin, which is a good indication of homogeneous catalyst/photosensitizer. Please see the following Figures:

Supplementary Figure 11. Photocatalytic CO₂ reduction. Plot of the initial rate of CO generation with respect to [FeTDHPP]. Photocatalytic CO₂ reduction in CO₂-saturated DMF solutions containing 100 μM F₂₀Ch, 20 mM BIH and with different initial [FeTDHPP] under red LED ($\lambda = 630$ nm, 110 mW/cm²) at 293 K.

Supplementary Figure 12. Photocatalytic CO₂ reduction. Photocatalytic CO₂ reduction in CO₂-saturated DMF solutions containing 20 μM FeTDHPP, 20 mM BIH and with different initial [F₂₀Ch] from 2 to 25 μM under red LED ($\lambda = 630$ nm, 110 mW/cm²) at 293 K (left); Plot of the initial rate of CO generation with respect to [F₂₀Ch] (right).

As to the relatively higher concentrations of catalyst and photosensitizer used in Figure 2e, one of our reasons was to highlight the scalability of the system, which was relatively less investigated in previous light-driven homogenous systems. Because deactivation of FeTDHPP limits the lifetime of the system, the system lasts longer at higher [FeTDHPP]. We have performed additional experiments and found that the activity of the system can be restored completely by addition of FeTDHPP. Please see the following Figure S36 and the statement added to the manuscript:

“Furthermore, addition of FeTDHPP to a photolysis system at 51 h completely restored the activity (Supplementary Fig. 36), which confirms that deactivation of FeTDHPP is a limiting factor in the lifetime of the system.”

Supplementary Figure 36. Photocatalytic CO₂ reduction. Photocatalytic systems containing 50 μM F₂₀Ch, 50 mM BIH, and 1 μM FeTDHPP under red LED ($\lambda = 630$ nm, 110 mW/cm²) at 293 K. FeTDHPP (1 μM) in DMF was added to the system under CO₂ at 51 h.

(2) Only CO production and TON were presented. The selectivity, CO₂ conversion, quantum efficiency, and solar-to-chemical efficiency should be provided as well.

We thank the referee for the comment. The quantum efficiency was presented in Supplementary Table 3. To make this clearer for the readers, we have added the following statement to the manuscript:

“With the per-fluorinated chlorin **F₂₀Ch**, the system achieves a high TON of 1790 ± 52 after 51 h of irradiation, an initial turnover frequency (TOF) up to $181 \pm 6 \text{ h}^{-1}$, and a quantum yield of $0.88 \pm 0.03\%$ at 630 nm (Supplementary Table 3).”

Supplementary Table 3. Quantum yields of CO production. Systems containing 50 μM F_xCh, 5 μM FeTDHPP, and 50 mM BIH in CO₂-saturated DMF under irradiation with red LEDs ($\lambda = 630 \text{ nm}$, 110 mW/cm^2) for 15 h.

PS	CO (μmol)	ΔP ($\times 10^{-4} \text{ W cm}^{-2}$)	ϕ_{CO} (%)
F ₀ Ch	1.33 ± 0.026	27.7	0.05 ± 0.001
F ₄ Ch	28.78 ± 1.451	35.3	0.91 ± 0.046
F ₁₂ Ch	4.45 ± 0.328	105.0	0.05 ± 0.003
F ₂₀ Ch	44.98 ± 1.739	57.0	0.88 ± 0.034

According to the literature, the activities of homogeneous photocatalytic reduction of CO₂ are generally evaluated by the following criteria: selectivity, quantum yield, turnover number (TON), and turnover frequency (TOF). Please see these review papers for example: *Chem. Rev.* 2019, 119, 2752-2875; *Acc. Chem. Res.* 2009, 42, 1983-1994; *ACS Catal.* 2017, 7, 3394-3409; *EnergyChem* 2020, 2, 100034. We regret that we did not include the selectivity of CO in the manuscript. We have investigated this further. There was no other gaseous product detected besides CO from experiments conducted under one atmosphere of CO₂. For experiments performed under low concentrations of CO₂, we did observe small amounts of H₂ from the systems, in which the selectivity was 95.6% (under 1% CO₂) and 97.7 % (under 5% CO₂). Regarding to the analysis of liquid products, we have tried to use ¹H NMR to detect formic acid and methanol. However, none of them could be observed. In this particular experiment, we were able to generate a large amount of CO (232 μmol) from the system. Our control experiments showed that even though the selectivity of formic acid or methanol was as low as 1%, we should be able to detect them by ¹H NMR. Thus, we conclude that there was no liquid product generated from the system.

In addition, we have performed photocatalytic experiments with high concentrations of FeTDHPP and F₂₀Ch. The results showed that the amounts of CO generated were very close to the theoretical yield of electrons that can be provided from BIH (based on two electrons per BIH molecule). Please see the following Figure S13. This evidence along with our GC and NMR measurements strongly suggest that the selectivity of CO is nearly 100%.

We have revised the manuscript by including the following statements to page 5:

“In all the red-light-driven experiments performed under an atmosphere of CO₂, no other gaseous product was observed by GC. Analysis of the liquid phase by ¹H NMR showed no detection of formic acid and methanol. To study the selectivity of the system further, we found that the amounts of CO generated were near the theoretical maximum value (Supplementary Fig. 13) of the electron donor BIH (based on two electrons per BIH molecule), which suggests that the selectivity of CO is nearly 100%.”

“Indeed, we observed slower rates of CO generation from mixtures at lower concentrations of CO₂ (Fig. 2f). However, its ability to function at low CO₂ contents (down to 1%) with high selectivities (97.7% under 5% CO₂ and 95.6% under 1% CO₂) was impressive.”

Supplementary Figure 13. Photocatalytic CO₂ reduction. Time profiles of photocatalytic CO₂ reduction in CO₂-saturated DMF solutions containing 100 μM FeTDHPP, 100 μM F₂₀Ch and 20 mM BIH under red LED ($\lambda = 630$ nm, 110 mW/cm²) at 293 K. Error bars denote standard deviations based on at least three separated runs. We have revised the manuscript by including the following statements:

(3) All the products should be analyzed. It seems that the liquid was not analyzed at all. HCOOH and other chemicals might be produced.

We thank the referee for the comment. We have investigated this further and found no HCOOH or other liquid products generated. Please see our reply to your comment (2).

(4) What's the light intensity?

The light intensity has been presented in the captions for all figures and tables in the MS and SI.

(5) Control experiments (activity tests, GC-MS measurements, CV analysis, etc.) were performed under N₂. N₂ is not really inert. Ar or He should be employed instead.

The carrier gas of GC-MS measurements was He. We have performed the control experiments for activity tests and CV measurements under Ar. The results were identical to those carried out under N₂. Please see the following table and figure:

Supplementary Table 6. Control experiments for photocatalytic CO₂ reduction^[a]

Entry	Condition	CO (μmol)
1	No PS	0
2	No catalyst	trace
3	No BIH	trace
4	Under Ar instead of CO ₂	0
5	No irradiation	0

[a] Standard conditions: a 5 mL CO₂-saturated DMF solution containing F₂₀Ch (50 μM), FeTDHPP (1 μM) and BIH (50 mM) was irradiated using red LED (λ = 630 nm, 110 mW/cm²) under a CO₂ atmosphere for 27 h.

Supplementary Figure 17. Electrochemical study. Cyclic voltammograms of 0.25 mM F₀Ch (black), 0.5 mM F₄Ch (green), 0.5 mM F₁₂Ch (blue), and 0.5 mM F₂₀Ch (red) in DMF containing 0.1 M TBAPF₆ under Ar (solid) or under N₂ (short dot) at a scan rate of 0.1 V·s⁻¹.

(6) A scheme illustrating the entire CO₂ reduction process (cycle) should be provided as a replacement of Fig. 3.

We thank the referee for the comment. We have performed further experiments to come up with a more complete mechanistic scheme for CO₂RR. Please see the following paragraphs and figures added to the manuscript and SI:

“Previous studies showed that BIH functioned as a $2e^-/1H^+$ donor^{59,60}. To generate the BI-radical (which donates the second e^-), deprotonation of the BIH-radical cation by a base such as triethylamine (TEA) was found to be necessary in ACN (Supplementary Table 11). However, there are several photocatalytic studies reported in DMF without TEA or additional bases when using BIH as the electron donor^{53,61,62}. To investigate this, we performed CV studies for BIH in DMF and ACN (Supplementary Figs. 46-47). In contrast to the spectrum in ACN, the CV in DMF showed appearance of a reduction wave at ~ -1.6 V vs SCE, corresponding to generation of the BI-radical. This result suggests that deprotonation of the BIH-radical cation is more favorable in DMF than in ACN. However, addition of TEA to the system was found to improve the activity (Supplementary Fig. 48 and Supplementary Table 11), exhibiting a TON_{CO} of 2132 in 27 h and an initial TOF_{CO} of $584 h^{-1}$. In the overall reactions, BI^+ and OH^- were produced in generation of the $2e^-/2H^+$ reduced chromophores and in CO_2 reduction (Eq. 1 and 2).

The photocatalytic CO_2 reduction mechanism by FeTDHPP has been extensively investigated by Robert and co-workers⁶³⁻⁶⁵. UV-vis studies showed generation of the corresponding Fe(II) and Fe(I) species at the early stage of photolysis (Supplementary Figs. 49-52). The Fe(I) species was found to decrease during CO production, which indicates a catalytic cycle consistent with previous reports (Fig. 4)^{52,64,65}. No electrostatic interaction was found between the chlorin and FeTDHPP by UV-vis studies (Supplementary Fig. 53), which suggests electron transfer from the chromophore to the catalyst follows an outer-sphere mechanism.”

Fig. 4 | Proposed mechanism for the red-light-driven CO_2RR .

Supplementary Figure 46. Electrochemical study. Cyclic voltammograms of 5 mM BIH in MeCN (left) or DMF (right) containing 0.1 M TBAPF₆ under Ar at a scan rate of 0.1 V·s⁻¹

Supplementary Figure 47. Electrochemical study. Cyclic voltammograms of 5 mM BIH in MeCN (bottom) or DMF (top) containing 0.1 M TBAPF₆ under Ar at a scan rate of 0.1 V·s⁻¹

Supplementary Figure 48. Photocatalytic CO₂ reduction. Time profiles of photocatalytic CO₂ reduction in CO₂-saturated DMF (black) or DMF:TEA = 4 :1 (red) solutions containing 50 μM F₂₀Ch, 1 μM FeTDHPP and 50 mM BIH under red LED (λ = 630 nm, 110 mW/cm²) at 293 K.

Supplementary Figure 49: UV-vis absorption spectra. Systems containing 10 μM F_0Ch , 10 μM FeTDHPP and 10 mM BIH in DMF under CO_2 upon irradiation with red LED light ($\lambda = 630 \text{ nm}$, 110 mW/cm^2) in a quartz cuvette (10-mm path length). Irradiation time ranging from 0 to 50 min (a) and from 50 min to 17 h (b).

Supplementary Figure 50: UV-vis absorption spectra. Systems containing 10 μM F_4Ch , 10 μM FeTDHPP and 10 mM BIH in DMF under CO_2 upon irradiation with red LED light ($\lambda = 630 \text{ nm}$, 110 mW/cm^2) in a quartz cuvette (10-mm path length). Irradiation time ranging from 0 to 2 min (a) and from 2 min to 20 min (b) and from 20 min to 13.5 h.

Supplementary Figure 51: UV-vis absorption spectra. Systems containing 10 μM F_{12}Ch , 10 μM FeTDHPP and 10 mM BIH in DMF under CO_2 upon irradiation with red LED light ($\lambda = 630 \text{ nm}$, 110 mW/cm^2) in a quartz cuvette (10-mm path length). Irradiation time ranging from 0 to 5 min (a) and from 5 min to 2 h (b) and from 2 h to 10 h (c).

Supplementary Figure 52: UV-vis absorption spectra. Systems containing 10 μM Fe_{20}Ch , 10 μM FeTDHPP and 10 mM BIH in DMF under CO_2 upon irradiation with red LED light ($\lambda = 630 \text{ nm}$, 110 mW/cm^2) in a quartz cuvette (10-mm path length). Irradiation time ranging from 0 to 30 min (a) and from 30 min to 2 h (b) and from 2 h to 5 h (c).

Reviewer #3:

This study by Han and co-workers is a very interesting body of work. While the overall TONs are impressive at first glance the large excess of BIH used and the limited mechanistic studies to truly understand the PCET chemistry occurring in this system are found lacking. The content is well written and the science presented is of reasonable quality however there is still many questions to be answered before publication in Nature, mostly concerning the role of BIH wrt PCET mechanisms with both photosensitizer and catalyst.

We would like to thank Reviewer 3 for his/her insightful reading of our manuscript and for the helpful comments and suggestions. Each point is addressed below.

Major revisions:

(1) I am somewhat surprised by the authors photocatalysis reaction conditions, e.g. using 5 mL DMF, 50mM BIH (5x10⁴ equivalents SED wrt catalyst), 1 μM FeTDHPP catalyst, and 50 μM photosensitizer. Did the authors not add any base, e.g. TEA, to their DMF solvent? The chemistry of BIH as a SED is fairly well established now. Single electron oxidation generates the BIH-radical cation. It is necessary to deprotonate this intermediate (typically with TEA) to make the BI-radical available for a second DARK electron transfer step to fully activate the catalyst. Thus, upon single photon absorption BIH donates a total of two-electrons (important for accurate QY calculations) and the TEAH-cation may provide a proton for C-OH bond cleavage at the catalyst for CO evolution. So not adding any TEA or alternative base to the catalysis system restricts this chemistry to the BIH-radical cation intermediate. This chemistry is further complicated by the fact that the Fe(III) porphyrin catalyst requires a 3e activation to access its Fe(0) active state, thus consuming 3e for initially activation and 2e for each propagated catalytic cycle once activated. At high TONs, the initial 3e activation can

likely be neglected. Interestingly, where the authors report QY eqns. they assume (not explicitly) that each cycle requires just a single photon (i.e. the numerator in eq. 1 page SI-5 is not multiplied by 2). This therefore assumes that their BIH is donating 2e per photon. The 2e donating properties of BIH are only established in the presence of TEA to generate the deprotonated BI-radical intermediate; unless there is a H-atom transfer step from the BIH-radical cation. This opens upon the question of H-atom transfer from the BIH-radical cation to the activated catalyst to potentially generate a metal-hydride intermediate (this requires appropriate BDFEs for exergonic HAT chemistry). However, such HAT chemistry initiated by the BIH-radical cation generally results in selective CO₂ insertion at the M-H bond and formate as the primary product of 2e induced CO₂ reduction. Thus I am at a loss as to the role of BIH in the authors catalytic system wrt propagation of the FDHTPP catalyst.

We thank the referee for bringing up several good points in this comment. We have performed further experiments to investigate the role of BIH. Indeed, deprotonation of the BIH-radical cation by TEA or other bases to generate the BI-radical has been studied extensively in ACN. However, there are several photocatalytic studies reported without using TEA, TEOA or other bases when using BIH as the electron donor. For example: *J. Am. Chem. Soc.* **2016**, *138*, 9413-9416; *J. Am. Chem. Soc.* **2016**, *138*, 13818-13821; *Nat. Commun.* **2021**, *12*, 1835. Please see the following figures for example.

Figure 4. CO (●) and H₂ (■) TONs as functions of time during irradiation of a CO₂-saturated DMF solution containing (a) 0.05 mM or (b) 0.005 mM **2**, 0.02 mM purpurin, and 0.1 M BIH.

Figure from *J. Am. Chem. Soc.* **2016**, *138*, 9413-9416

Figure 4. Profound influence on photocatalytic CO₂ reduction by triethanolamine (a) photocatalytic reactions under 100% and 1% CO₂ atmosphere with/without TEOA. (b) Linear relationship between the initial rate of CO formation and the concentration of the CO₂-capturing complex Ru–Re–OC(O)OC₂H₄NR₂.

Figure from *J. Am. Chem. Soc.* **2016**, *138*, 13818-13821

We noted that all these studies were in DMF instead of ACN. We have performed CV experiments for BIH in both DMF and ACN. We found that no reduction wave is corresponding to the BI-radical (at ~ -1.6 V vs SCE) in ACN, which is consistent with previous study that the BI-radical has to be generated by deprotonation of the BIH-radical cation by a base. However, we were able to observe the reduction wave of BI-radical in DMF (by first scanning towards positive potential). Please see the following figures. This result suggests that the deprotonation of BIH-radical in DMF is much easier than in ACN.

Supplementary Figure 46. Electrochemical study. Cyclic voltammograms of 5 mM BIH in MeCN (left) or DMF (right) containing 0.1 M TBAPF₆ under Ar at a scan rate of 0.1 V·s⁻¹

Supplementary Figure 47. Electrochemical study. Cyclic voltammograms of 5 mM BIH in MeCN (bottom) or DMF (top) containing 0.1 M TBAPF₆ under Ar at a scan rate of 0.1 V·s⁻¹

To study whether there is H-atom transfer from the BIH-radical cation to the activated catalyst, which might lead to generation of formate, we have re-analyzed the selectivity of the catalytic system. There was no other gaseous product detected besides CO from experiments conducted under one atmosphere of CO₂. For experiments performed under low concentrations of CO₂, we did observe small amounts of H₂ from the systems,

in which the selectivity was 95.6% (under 1% CO₂) and was 97.7 % (under 5% CO₂). Regarding to the analysis of liquid products, we have tried to use ¹H NMR to detect formic acid and methanol. However, none of them could be observed. In this particular experiment, we were able to generate a large amount of CO (232 μmol) from the system. Our control experiments showed that even though the selectivity of formic acid or methanol was as low as 1%, we should be able to detect them by ¹H NMR. Thus, we conclude that there was no liquid product generated from the system.

In addition, we have performed photocatalytic experiments with high concentrations of FeTDHPP and F₂₀Ch. The results showed that the amounts of CO generated were very close to the theoretical yield of electron that can be provided from BIH (based on two electrons per BIH molecule). Please see the following Figure S13. This evidence along with our GC and NMR measurements strongly suggest that the selectivity of CO is nearly 100%. These results suggest that H-atom transfer from the BIH-radical cation to the Fe catalyst is unlikely.

We have revised the manuscript by including the following statements to page 5:

“In all the red-light-driven experiments performed under an atmosphere of CO₂, no other gaseous product was observed by GC. Analysis of the liquid phase by ¹H NMR showed no detection of formic acid and methanol. To study the selectivity of the system further, we found that the amounts of CO generated were near the theoretical maximum value (Supplementary Fig. 13) of the electron donor BIH (based on two electrons per BIH molecule), which suggests that the selectivity of CO is nearly 100%.”

“Indeed, we observed slower rates of CO generation from mixtures at lower concentrations of CO₂ (Fig. 2f). However, its ability to function at low CO₂ contents (down to 1%) with high selectivities (97.7% under 5% CO₂ and 95.6% under 1% CO₂) was impressive.”

Supplementary Figure 13. Photocatalytic CO₂ reduction. Time profiles of photocatalytic CO₂ reduction in CO₂-saturated DMF solutions containing 100 μM FeTDHPP, 100 μM F₂₀Ch and 20 mM BIH under red LED ($\lambda = 630$ nm, 110 mW/cm²) at 293 K. Error bars denote standard deviations based on at least three separated runs. We have revised the manuscript by including the following statements:

(2) The authors mechanistic studies do suggest that the BIH-radical cation is reacting with their chlorins photosensitizers, via visible absorption evidence of 2e⁻/2H⁺ reduced chlorinphlorins. Yet, as discussed above, BIH is known to be just a 2E⁻/1H⁺ donor, so just a single proton donor, and even then only when in the presence of a strong base such as TEA. Keeping in mind that residual H₂O in an insufficient base for BIH-radical cation deprotonation in DMF, can the authors address the lack of stoichiometry here and how such chemistry may occur in the absence of any added base? While I appreciate the authors focus is on their photosensitizers, even though there are still question here wrt the PCET chemistry of BIH (and oxidized versions thereof) with chlorins, I believe this is the first report of photocatalytic CO₂ reduction by the FDHTPP catalyst with BIH and this chemistry which is very critical to this study is not reported (to the best of my knowledge) nor is it addressed in any shape or form by the authors here.

We thank the referee for the comment. As described above (our reply to your comment 1), BIH functions as a 2e⁻/1H⁺ donor in DMF (may be with the aid of residual H₂O). In fact, we have previously reported a few papers for photocatalytic CO₂ reduction using FeTDHPP and BIH (*Nat. Commun.* **2021**, *12*, 1835; *Nat. Commun.* **2023**, *14*, 1087). In these studies, we focused on the developments of anthraquinone-based photosensitizers. However, we have not addressed this question in these papers. We appreciate your valuable comments on this. Regarding the stoichiometry, we have revised our mechanistic scheme and have added the following statements to the manuscript to make this point clearer for readers:

“Previous studies showed that BIH functioned as a 2e⁻/1H⁺ donor^{59,60}. To generate the BI-radical (which donates the second e⁻), deprotonation of the BIH-radical cation by a base such as triethylamine (TEA) was found to be necessary in ACN (Supplementary Table 11). However, there are several photocatalytic studies reported in DMF without TEA or additional bases when using BIH as the electron donor^{53,61,62}. To investigate this, we performed CV studies for BIH in DMF and ACN (Supplementary Figs. 46-47). In contrast to the spectrum in ACN, the CV in DMF showed appearance of a reduction wave at ~ -1.6 V vs SCE, corresponding to generation of the BI-radical. This result suggests that deprotonation of the BIH-radical cation is more favorable in DMF than in ACN. However, addition of TEA to the system was found to improve the activity (Supplementary Fig. 48 and Supplementary Table 11), exhibiting a TON_{CO} of 2132 in 27 h and an initial TOF_{CO} of 584 h⁻¹. In the overall reactions, BI⁺ and OH⁻ were produced in generation of the 2e⁻/2H⁺ reduced chromophores and in CO₂ reduction (Eq. 1 and 2).

The photocatalytic CO₂ reduction mechanism by FeTDHPP has been extensively investigated by Robert and co-workers⁶³⁻⁶⁵. UV-vis studies showed generation of the corresponding Fe(II) and Fe(I) species at the early stage of photolysis (Supplementary Figs. 49-52). The Fe(I) species was found to decrease during CO production, which indicates a catalytic cycle consistent with previous reports (Fig. 4)^{52,64,65}. No electrostatic interaction was found between the chlorin and FeTDHPP by UV-vis studies (Supplementary Fig. 53), which suggests electron transfer from the chromophore to the catalyst follows an outer-sphere mechanism.”

Fig. 4 | Proposed mechanism for the red-light-driven CO₂RR.

(3) The very large excess of BIH used (4 orders of magnitude excess) is also never addressed. At least when utilized in the presence of TEA, BIH is often the limiting reagent wrt TON. Although there are some high TONs here report wrt related literature the relative ratios of BIH are not taken into account here. In this respect the TONs are actually relatively low. I strongly suggest that the authors provide some clarity here, at least for comparison to the bulk literature, by providing reference experiments with photocatalysis conducted in DMF:TEA 4:1 this would very much benefit the readers. Note, TEOA is to be avoided as it is not basic enough to fully deprotonate the BIH-radical cation (see Fujita and co-workers *J. Am. Chem. Soc.* 2020, 142, 2413–2428)

We thank the referee for the comment. As mentioned previously in our replies to your comments 1 and 2, the BIH can be completely consumed and indeed is the limiting reagent wrt TON.

In fact, there are a lot of studies using excess amounts of BIH ($10^3 \sim 10^4$ equivalents of BIH wrt catalyst). Please see the following examples:

ACS Catal. **2023**, *13*, 5979–5985;

Chem. Commun. **2023**, *59*, 10741-10744;

J. Am. Chem. Soc. **2023**, *145*, 23196-23204;

J. Am. Chem. Soc. **2019**, *141*, 20309-20317;

J. Am. Chem. Soc. **2017**, *139*, 6538-6541;

J. Am. Chem. Soc. **2020**, *142*, 10261-10266.

We have also performed the experiments in 4:1 DMF:TEA. Indeed, adding TEA to the system significantly increased the activity. Please see the following figure and table. Please also see our reply to your comment 2 for added statement.

Supplementary Figure 48. Photocatalytic CO₂ reduction. Time profiles of photocatalytic CO₂ reduction in CO₂-saturated DMF (black) or DMF:TEA = 4 :1 (red) solutions containing 50 μM F₂₀Ch, 1 μM FeTDHPP and 50 mM BIH under red LED ($\lambda = 630 \text{ nm}$, 110 mW/cm^2) at 293 K.

Supplementary Table 11. Molecular systems for CO₂ reduction with TEA additive.

System	Major product	Solvent	Sacrificial donor	TON (time)	TOF (h ⁻¹)	Light source (nm)	Ref
F ₂₀ Ch(50 μM) + FeTDHPP (1 μM)	CO	20%TEA/DMF	BIH (50 mM)	2132 (27 h)	584	630	This work
[TATA]PF ₆ (0.2 mM) + [Co(qpy)(OH ₂) ₂] ²⁺ (2 μM)	CO HCOOH	0.15 M TEA/MeCN	BIH (100 mM)	182 (8 h) 419 (8 h)	-	450	26
[Ru(phen) ₃] ²⁺ (0.2 mM) + [Co ₂ biqpy] ⁴⁺ (50 μM)	HCOOH	20%TEA/MeCN	BIH (25 mM)	386 (23 h)	-	460	27
Re(PyNHC-PhCF ₃)(CO) ₃ Br (100 uM) + fac-Ir(ppy) ₃ (100 uM)	CO	5%TEA/MeCN	BIH (10 mM)	51 (4 h)	12.8	a solar simulator (AM1.5 filter)	28
CuBCP (500 uM) + CoFPC (0.5 μM)	CO	CH ₃ CN/TEA = 5	BIH (20 mM)	9185 (-)	-	425	29
IrQPY (100 uM) + CoTPA (100 uM)	CO	2.5%TEA/MeCN	BIH (80 mM)	391 ± 7(4 h)	-	450 ± 5	30
CuPYBCP (100 uM) + CoPYN5 (100 uM)	CO	1 M TEA/MeCN	BIH (20 mM)	338.0 ± 43.5 (2 h)	263.7 ± 16.1	425	31
Ir(ppy) ₃ (100 uM) + Au-NHC (2 ^{Cl}) (0.1 uM)	CO	5%TEA/MeCN	BIH (20 mM)	270 (2 h)	28.2	380-750	32
CuPPBCP (500 uM) + CoTcPc (0.5 uM)	CO	CH ₃ CN/TEA=5	BIH (50 mM)	11800 ± 1400 (4 h)	11.8 ± 1.4	425	33

Minor:

(1) Abstract “However, a molecular system that mimics such function has not been demonstrated in non-noble-metal catalysis” This is not true as per the article by Julia Weinstein and co-workers, *Inorg. Chem.* 2022, 61, 34, 13281–13292. The authors cite this article further below (#21). Although its TON is low it still is fact and makes the authors claim here false.

We thank the referee for the comment. We have modified the sentence to be “However, a molecular system that mimics such function is extremely rare in non-noble-metal catalysis”.

(2) P2 L21-23 “Because of the large energy barriers associated with the activation of CO₂, high energetic light is generally required in driving such transformation.” This statement is very crude and represents a poor interpretation of the recent literature wrt the general photocatalytic CO₂ reduction literature. Whether using noble or non-noble metals there is a growing literature using red light for photocatalytic CO₂ reduction, as the authors have pointed out and cited in their following paragraph. Therefore they are contradicting themselves in stating that high energy light is a requirement. Recent literature they have cited demonstrates that this is not the case. High energy light is not required – it has simply been the most commonly utilized region

of the spectrum in photocatalytic CO₂RR based upon earlier catalytic design; light restrictions of prior research in this field should not be interpreted as fundamental thermodynamic requirements for the activation of CO₂ toward CO production. The growing literature of red-light induced CO₂ reduction photocatalysis simply demonstrates a poor design of previous photocatalytic systems reported in the literature-to date which are not capable of harvesting a lower energy red-light input for CO₂ reduction.

We thank the referee for the comment. We have deleted this incorrect statement from the manuscript.

Reviewer #4:

The manuscript entitled "Fluorinated chlorin chromophores for red-light-driven CO₂ reduction" is an interesting paper proceeding in the presence of an iron based complex FeTDHPP (that is in precious-metal free conditions).

My main question concerns the precautions taken by the authors to ensure that the system is not contaminated by impurities of precious metals (but also copper, nickel...), which could possibly be wholly or partly responsible for the CO₂ reduction.

We would like to thank Reviewer 4 for his/her insightful reading of our manuscript and for the helpful comments and suggestions.

For example, what is the commercial purity (transition metal based) of the iron salts used in the synthesis of FeTDHPP (98.6.. or 99.99..) ? I did not find the information neither in the paper nor in the ref 2 indicated for its synthesis. I have the same question regarding the purity of the K₂CO₃ used to prepare FxChs from FxTPP. What is its commercial purity? (It's not uncommon for traces of transition metals to be present in such a base). Even if in this case I think there's no problem because the TON of F12TPP is more or less comparable to that of F12CH (Fig 2a), having the information would be interesting. Also, I understand that the commercial solvent (DMF) is not purified/distilled before using. Is it the case?

Overall, I don't think there's any impurity problem, but at least one test under perfect conditions with extremely pure reagents (including FeTDHPP), would be more reassuring and would reinforce these interesting results.

We thank the referee for the comment.

The purity of Fe(NO₃)₃•9H₂O was ≥ 98.5%.

The purity of K₂CO₃ was ≥ 99.0%.

The commercial DMF (99.8%) was distilled prior to use.

The DMF was distilled.

Purities of ALL chemicals used in our study have been added to the SI. Please see the following information from the revised SI:

“DMF (Energy chemical, 99.8%, distilled, extra dry with molecular sieves, water \leq 50 ppm (by K.F.)) was purchased from Anhui Zesheng Technology Co., Ltd (Anhui, China); Anhydrous K_2CO_3 (GREAGENT, \geq 99.0%) was purchased from Guangzhou beier biological Technology Co., Ltd (Guangzhou, China); *p*-Toluenesulfonyl hydrazide (Macklin, 98%) was purchased from Guangzhou Sopo biological Technology Co., Ltd (Guangzhou, China); Dry pyridine (Acseal, 99.5%, with molecular sieves, water \leq 50 ppm (by K.F.)) was purchased from Shanghai Jizhi Biochemical Technology Co., Ltd (Shanghai, China); 2,3-Dichloro-5,6-dicyano-benzoquinone (DDQ, Energy chemical 98%) was purchased from Anhui Zesheng Technology Co., Ltd (Anhui, China); Benzaldehyde (innocem, 98%); 2-Fluorobenzaldehyde (BIDE, 98%); 2,4,6-Trifluorobenzaldehyde (BIDE, 98%); Pentafluorobenzaldehyde (macklin, 98%); 2,6-Dimethoxybenzaldehyde (BIDE, 98%); Boron trifluoride diethyl etherate (Aladdin, BF_3 : 46.5%); Pyrrole (Energy chemical, 99%); $Co(NO_3)_3 \cdot 6H_2O$ (damas-beta, 99.0%); $Fe(NO_3)_3 \cdot 9H_2O$ (Guangzhaou, \geq 98.5%); $Cu(NO_3)_2 \cdot 9H_2O$ (Kermel, 99.0~102.0%) ; $Ni(NO_3)_2 \cdot 6H_2O$ (Xiya, 99%); $RuCl_3 \cdot xH_2O$ (Aladdin, 35.0-42.0% Ru basis); Dichloromethane (WOHUA-CHEMICAL, 99.5%); Petroleum ether (WOHUA-CHEMICAL, 99.5%); Ethyl acetate (WOHUA-CHEMICAL, 99.5%).”

We have also performed C/H/N analysis for all the photosensitizers, catalyst, and electron donor in our study. And the results suggest they are pure. Please see the results listed below. We have included these results to the SI.

F₀TPP: Anal. Calcd. For $C_{44}H_{30}N_4 \cdot 0.8H_2O$: C, 84.00; H, 5.06; N, 8.91; found: C, 84.00; H, 5.07; N, 8.81.

F₄TPP: Anal. Calcd. For $C_{44}H_{26}F_4N_4 \cdot 0.3H_2O$: C, 76.36; H, 3.87; N, 8.10; found: C, 76.49; H, 4.18; N, 8.06.

F₁₂TPP: Anal. Calcd. For $C_{44}H_{18}F_{12}N_4 \cdot 0.3H_2O$: C, 63.21; H, 2.24; N, 6.70; found: C, 63.49; H, 2.59; N, 6.77.

F₂₀TPP: Anal. Calcd. For $C_{44}H_{10}F_{20}N_4 \cdot 0.7H_2O \cdot 0.1C_6H_{14}$ (0.1 Hex): C, 53.80; H, 1.30; N, 5.63; found: C, 53.85; H, 1.31; N, 5.81.

F₀Ch: Anal. Calcd. For $C_{44}H_{32}N_4 \cdot 0.4H_2O$: C, 84.70; H, 5.30; N, 8.98; found: C, 84.77; H, 5.47; N, 8.96.

F₄Ch: Anal. Calcd. For $C_{44}H_{28}F_4N_4 \cdot 0.4H_2O$: C, 75.94; H, 4.17; N, 8.05; found: C, 75.67; H, 4.45; N, 7.97.

F₁₂Ch: Anal. Calcd. For $C_{44}H_{20}F_{12}N_4$: C, 63.47; H, 2.42; N, 6.73; found: C, 63.50; H, 2.60; N, 6.77.

F₂₀Ch: Anal. Calcd. For C₄₄H₁₂F₂₀N₄: C, 54.12; H, 1.24; N, 5.74; found: C, 54.42; H, 1.54; N, 5.94.

F₂₀BC: Anal. Calcd. For C₄₄H₁₄F₂₀N₄•0.3C₆H₁₄ (0.3 Hex): C, 54.77; H, 1.83; N, 5.58; found: C, 54.79; H, 1.91; N, 5.61.

FeTDHPP: Anal. Calcd. for C₄₄H₂₈ClFeN₄O₈•1.7H₂O•C₆H₁₄(Hex): C, 63.29; H, 4.82; N, 5.90; found: C, 63.39; H, 5.11; N, 5.70.

BIH: Anal. Calcd. For C₁₅H₁₆N₂: C, 80.32; H, 7.19; N, 12.49; found: C, 80.20; H, 7.26; N, 12.42.

We have also conducted photocatalytic experiments in freshly distilled DMF with chemicals that passed the EA tests. The results were identical to the ones we presented previously. Please see the following figure:

Photocatalytic CO₂ reduction in CO₂-saturated DMF containing 50 μM F₂₀Ch, 1 μM FeTDHPP, and 50 mM BIH under red LED ($\lambda = 630$ nm, 110 mW/cm²) at 293 K for 51 h. Error bars denote standard deviations, based on at least three separated three runs.

In addition, we have performed several control experiments to rule out potential contaminants from inorganic salts. Please see the following table. Inorganic salts (Fe³⁺, Cu²⁺, Ni²⁺, Co³⁺, Ru³⁺) with or without adding the TDHPP ligand ALL produced no or negligible amount of CO as compared with the experiment using FeTDHPP as the catalyst.

Supplementary Table 7. Control experiments with inorganic salts.

Entry	Catalyst	CO (μmol)	H ₂ (μmol)
1	FeTDHPP (1 μM)	8.2 \pm 0.63	0
2	Fe(NO ₃) ₃ (10 μM)	0.1	trace
3	Cu(NO ₃) ₂ (10 μM)	0	0.17
4	Ni(NO ₃) ₂ (10 μM)	0.14	trace
5	Co(NO ₃) ₃ (10 μM)	0	trace
6	RuCl ₃ (10 μM)	0.15	trace
7	Fe(NO ₃) ₃ (1 μM) + TDHPP (1 μM)	0.12	0
8	Cu(NO ₃) ₂ (1 μM) + TDHPP (1 μM)	0.11	0
9	Ni(NO ₃) ₂ (1 μM) + TDHPP (1 μM)	0.13	0
10	Co(NO ₃) ₃ (1 μM) + TDHPP (1 μM)	trace	0
11	RuCl ₃ (1 μM) + TDHPP (1 μM)	0.11	0

Reaction conditions: a 5 mL CO₂-saturated DMF solution containing F₂₀Ch (50 μM), BIH (50 mM) and metal catalyst was irradiated using red LED ($\lambda = 630 \text{ nm}$, 110 mW/cm²) under a CO₂ atmosphere for 27 h.

We have added the following description to the manuscript:

“To rule out potential metal contaminants, inorganic salts (Fe³⁺, Cu²⁺, Ni²⁺, Co³⁺, Ru³⁺) with or without TDHPP ligand all produced no or negligible amount of CO as compared with the experiment using FeTDHPP as the catalyst (Supplementary Table 7).”

Reviewers' Comments:

Reviewer #1:

Remarks to the Author:

As stated during the first review, I find that the work is well done, the experiments have been well performed with a range of techniques being employed, including photophysical and electrochemical analyses. A substantial amount of data is presented and the paper is well written. The isotopic labeling experiments are convincing, together with all the other control experiments. The conclusions are well supported by the data presented. This is a very nice study and I recommended publication after addressing some important points.

After seeing the revisions, I am pleased with the authors' answers and the effort they put in addressing the reviewers' comments. I am not asking it in this case, but as a suggestion for C/H/N elemental analysis – instead of fitting fractions of solvent molecules, it would be advisable to dry better the product and rerun the analysis.

In conclusion, I recommend publication of this work.

Minor (but important) correction:

- Line 60 in the manuscript: 'All chlorins exhibit an intense **sorbet** (B) **bond** (from 405 to 420 nm).' to be corrected to 'All chlorins exhibit an intense **Soret** (B) **band** (from 405 to 420 nm).'

Reviewer #2:

Remarks to the Author:

After reading the responses from the authors and the revised manuscript, I still think the poor stability (within 20 hours) greatly diminishes the significance of this work. In addition, the authors neither provided the solar-to-chemical efficiency nor described the reason for avoiding providing the result.

Reviewer #3:

Remarks to the Author:

I commend the authors for their comprehensive response to reviewers and forthwith recommend this article for publication.

I have one minor revision wrt the authors discussion of BIH voltammetry. They refer to this experimental result in acetonitrile as a "spectrum". Voltammetry data should never be referred to as spectra - there is no light absorption involved. Please refer to this data plot as a "voltamogram".

Reviewer #4:

Remarks to the Author:

If I understand their info sup correctly, the authors prepared FeTDHPP from reference 2, i.e. using FeBr₂. So it's not the purity of FeNO₃ (available in the info sup) that I'd like to know, but that of FeBr₂.

Is the purity of commercial FeBr₂ corresponds to 99.995% metal basis (easy to buy from several suppliers)?

If not, I'd like to see at least one general test (if possible, more) with FeTDHPP prepared from this extremely pure FeBr₂ iron precursor.

In addition, I'd like to see a test with a palladium salt, in Table 7

Reviewer #1:

As stated during the first review, I find that the work is well done, the experiments have been well performed with a range of techniques being employed, including photophysical and electrochemical analyses. A substantial amount of data is presented and the paper is well written. The isotopic labeling experiments are convincing, together with all the other control experiments. The conclusions are well supported by the data presented. This is a very nice study and I recommended publication after addressing some important points.

After seeing the revisions, I am pleased with the authors' answers and the effort they put in addressing the reviewers' comments. I am not asking it in this case, but as a suggestion for C/H/N elemental analysis – instead of fitting fractions of solvent molecules, it would be advisable to dry better the product and rerun the analysis.

In conclusion, I recommend publication of this work.

We would like to thank Reviewer 1 for his/her helpful comments and critical reading of our manuscript.

In fact, all samples have been dried under a high vacuum over one week prior to C/H/N analysis. However, we don't have the apparatus to completely seal the samples when transferring them to the institute for measurements. The weather in Guangzhou was very humid in the past few months, generally with humidity over 80% in the lab. Thus, a small amount of water absorbed by the sample during shipping or measurement was likely to happen. It is worth mentioning that the NMR spectra, HR-MS spectra, and control experiments (Supplementary Table 7) also confirmed the samples were pure.

Minor (but important) correction:

*- Line 60 in the manuscript: 'All chlorins exhibit an intense **sorbet (B) bond** (from 405 to 420 nm).' to be corrected to 'All chlorins exhibit an intense **Soret (B) band** (from 405 to 420 nm).'*

This was corrected.

Reviewer #2:

After reading the responses from the authors and the revised manuscript, I still think the poor stability (within 20 hours) greatly diminishes the significance of this work. In addition, the authors neither provided the solar-to-chemical efficiency nor described the reason for avoiding providing the result.

Regarding stability of the system, as we have described in the last response letter, our results showed that the deactivation of FeTDHPP is one of the main reasons that limits

the lifetime of the system: the activity of the system can be restored by adding FeTDHPP (Supplementary Fig. 37). We have also presented photocatalytic conditions to prolong the lifetime of the system over 240 hours (Fig 2d and 2e). Because the main focus of this paper is on the development of chromophores for red-light-driven CO₂ reduction, improving the stability of the catalyst to achieve more stable systems is certainly part of our goals in future studies.

For the half-reaction in photosynthetic homogeneous systems, solar-to-chemical efficiency (STC) is not commonly presented in the literature. Please see these review papers for example: *Chem. Rev.* 2019, 119, 2752-2875; *Acc. Chem. Res.* 2009, 42, 1983-1994; *ACS Catal.* 2017, 7, 3394-3409; *EnergyChem* 2020, 2, 100034. In these papers, STC data was not included in the evaluation criteria. A rationalization (we think) for this could be that the uses of sacrificial agents and solvent mixtures really complicate the calculation of energy conversion.

Reviewer #3:

I commend the authors for their comprehensive response to reviewers and forthwith recommend this article for publication.

I have one minor revision wrt the authors discussion of BIH voltammetry. They refer to this experimental result in acetone as a "spectrum". Voltammetry data should never be referred to as spectra - there is no light absorption involved. Please refer to this data plot as a "voltamogram".

We would like to thank Reviewer 3 for his/her helpful comments and critical reading of our manuscript. "Voltamogram" has been used to replace "Spectrum" for CV measurements.

Reviewer #4:

If I understand their info sup correctly, the authors prepared FeTDHPP from reference 2, i.e. using FeBr₂. So it's not the purity of FeNO₃ (available in the info sup) that I'd like to know, but that of FeBr₂. Is the purity of commercial FeBr₂ corresponds to 99.995% metal basis (easy to buy from several suppliers)? If not, I'd like to see at least one general test (if possible, more) with FeTDHPP prepared from this extremely pure FeBr₂ iron precursor.

We would like to thank Reviewer 4 for his/her helpful comments and critical reading of our manuscript.

We prepared the FeTDHPP following the procedure from reference 2 (*Science* 2012, 338, 90-94), but using FeCl₂•4H₂O as the iron precursor instead of FeBr₂. The purity of the FeCl₂•4H₂O was 99.5%-101.0%. This information has been added to the manuscript. We have also prepared the FeTDHPP using FeBr₂ (99.995% metals basis) as the

precursor. Photocatalytic experiments showed identical results for FeTDHPP synthesized from either $\text{FeCl}_2 \cdot 4\text{H}_2\text{O}$ or FeBr_2 , or using all chemicals that have passed the EA tests. Please see the following figure. We have added the figure to the SI and a sentence to the main text to describe the results:

“Photolysis performed using chemicals that passed the elemental analysis or using FeTDHPP synthesized from highly pure FeBr_2 all showed identical activity (Supplementary Fig. 15).”

Supplementary Figure 15. Photocatalytic CO_2 reduction. Photocatalytic CO_2 reduction in CO_2 -saturated DMF containing 50 μM F_{20}Ch , 1 μM FeTDHPP, and 50 mM BIH under red LED ($\lambda = 630 \text{ nm}$, 110 mW/cm^2) at 293 K for 51 h. Error bars denote standard deviations, based on at least three separated runs. Purity of $\text{FeCl}_2 \cdot 4\text{H}_2\text{O}$ is 99.5%-101.0%. EA denotes elemental analysis. Purity of FeBr_2 is 99.995%.

In addition, I'd like to see a test with a palladium salt, in Table 7

We have added experiments using $\text{Pd}(\text{OAc})_2$ (Eybridge, 98%) with or without adding the TDHPP ligand. Both experiments produced very small amounts of CO as compared with the experiment using FeTDHPP as the catalyst. $\text{Pd}(\text{OAc})_2$ without TDHPP added produced mainly H_2 . Please see the following revised Table S7.

Supplementary Table 7. Control experiments with inorganic salts.

Entry	Catalyst	CO (μmol)	H ₂ (μmol)
1	FeTDHPP (1 μM)	8.2 \pm 0.63	0
2	Fe(NO ₃) ₃ (10 μM)	0.1	trace
3	Cu(NO ₃) ₂ (10 μM)	0	0.17
4	Ni(NO ₃) ₂ (10 μM)	0.14	trace
5	Co(NO ₃) ₃ (10 μM)	0	trace
6	RuCl ₃ (10 μM)	0.15	trace
7	Pd(OAc)₂ (10 μM)	0.17	16.3
8	Fe(NO ₃) ₃ (1 μM) + TDHPP (1 μM)	0.12	0
9	Cu(NO ₃) ₂ (1 μM) + TDHPP (1 μM)	0.11	0
10	Ni(NO ₃) ₂ (1 μM) + TDHPP (1 μM)	0.13	0
11	Co(NO ₃) ₃ (1 μM) + TDHPP (1 μM)	trace	0
12	RuCl ₃ (1 μM) + TDHPP (1 μM)	0.11	0
13	Pd(OAc)₂ (1 μM) + TDHPP (1 μM)	0	trace

Reaction conditions: a 5 mL CO₂-saturated DMF solution containing F₂₀Ch (50 μM), BIH (50 mM) and metal catalyst was irradiated using red LED ($\lambda = 630 \text{ nm}$, 110 mW/cm²) under a CO₂ atmosphere for 27 h.

We have revised the manuscript as follows:

The sentence “To rule out potential metal contaminants, inorganic salts (Fe³⁺, Cu²⁺, Ni²⁺, Co³⁺, Ru³⁺) with or without TDHPP ligand all produced no or negligible amount of CO as compared with the experiment using FeTDHPP as the catalyst (Supplementary Table 7).” was replaced by “To rule out potential metal contaminants, inorganic salts (Fe³⁺, Cu²⁺, Ni²⁺, Co³⁺, Ru³⁺, **Pd²⁺**) with or without TDHPP ligand all produced no or negligible amount of CO as compared with the experiment using FeTDHPP as the catalyst (Supplementary Table 7).”

Reviewers' Comments:

Reviewer #4:

Remarks to the Author:

I'm satisfied with this third version and with the authors' answers to my questions (about purity of starting iron precursors)

I recommend the publication of this paper in Nature Communications